# WHERE DID THE REASONING GO WRONG? A BENCHMARK OF PUZZLE-BASED VISUAL TASKS WITH COT ERROR DETECTION

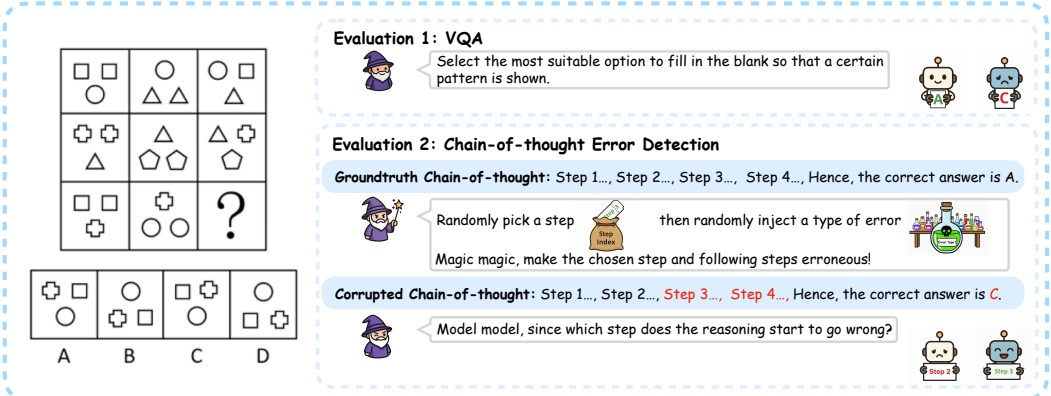

Figure 1: Overview of the proposed benchmark for multimodal reasoning, which aims to evaluate Multimodal LLMs on ($i$) solving visual puzzles, and ($ii$) their ability to detect where the reasoning goes wrong in erroneous reasoning.

## ABSTRACT

Multimodal large language models (MLLMs) have achieved remarkable progress on vision–language tasks, yet their reasoning processes remain sometimes unreliable. We introduce PRISM-Bench[1], a benchmark of puzzle-based visual challenges designed to evaluate not only whether models can solve problems, but how their reasoning unfolds. Unlike prior evaluations that measure only final-answer accuracy, PRISM-Bench introduces a diagnostic task: given a visual puzzle and a step-by-step chain-of-thought (CoT) containing exactly one error, models must identify the first incorrect step. This setting enables fine-grained assessment of logical consistency, error detection, and visual reasoning. The puzzles in PRISM-Bench require multi-step symbolic, geometric, and analogical reasoning, resisting shortcuts based on superficial pattern matching. Evaluations across state-of-the-art MLLMs reveal a persistent gap between fluent generation and faithful reasoning: models that produce plausible CoTs often fail to locate simple logical faults. By disentangling answer generation from reasoning verification, PRISM-Bench offers a sharper lens on multimodal reasoning competence and underscores the need for diagnostic evaluation protocols in the development of trustworthy MLLMs.

## 1 INTRODUCTION

Multimodal reasoning is central to human cognition. While recent Multimodal Large Language Models (MLLMs) such as GPT-o3 (OpenAI, 2025b), MiMo-VL-7B (Xiaomi, 2025), VL-Rethinker-7B (Wang et al., 2025) exhibit strong capabilities in perception and text generation, their capacity for reasoning over complex visual inputs remains underexplored. Most existing benchmarks probe

---

[1]Short for Puzzle Reasoning with In-Sequence Mistakes.

reasoning only through VQA-style tasks: a model is shown an image and a question, and evaluation reduces to checking the correctness of a single final answer. While effective for measuring end-to-end problem solving, this paradigm conflates perception, shallow pattern recognition, and reasoning into one metric. As a result, it offers limited insight into how models reason and where their reasoning may go wrong.

A key gap is the lack of benchmarks that explicitly evaluate reasoning fidelity. Some recent efforts (Hao et al., 2025; Yue et al., 2024b) have scaled up domains, filtered out text-only solvable samples, or introduced diagram-based mathematics and compositional puzzles. Yet, they still stop short of verifying the stepwise validity of model reasoning. This leaves open the question: can MLLMs not only solve visual problems, but also detect errors in reasoning processes?

To address this, we introduce PRISM-Bench, a benchmark that goes beyond answer accuracy. Each puzzle is paired with both a ground-truth chain of thought and a corrupted chain of thought. To construct the corrupted version, we randomly choose a step in the reasoning and rewrite that step and all subsequent steps so that they remain coherent but contain exactly one injected error. This guarantees that the first error occurs precisely at the selected step, while earlier steps remain valid. Models must then identify this point of failure; a task we call first-error detection. PRISM-Bench combines: 1) Challenging puzzle-based visual tasks that demand multi-step symbolic, geometric, and analogical reasoning, preventing shortcut solutions; 2) A dual evaluation protocol: ($i$) direct puzzle solving (final answer), and ($ii$) reasoning verification.

This dual setup disentangles generation from verification. Solving puzzles tests a model's ability to produce answers, while first-error detection probes whether it can audit reasoning faithfully. Evaluations across state-of-the-art MLLMs reveal a striking gap: models often produce fluent yet flawed explanations, failing to locate even simple logical errors. Furthermore, performance across the two tracks is often uncorrelated, suggesting that success in answer prediction does not imply genuine stepwise understanding.

In summary, our contributions are threefold: ($i$) **Benchmark design**: a suite of puzzle-based visual reasoning tasks requiring multi-step symbolic, geometric, and analogical inference; ($ii$) **Dual evaluation protocol**: complementary tracks for final-answer prediction and chain-of-thought error detection, enabling fine-grained diagnostic analysis; ($iii$) **Comprehensive evaluation**: an empirical study across frontier MLLMs, revealing persistent gaps between fluent reasoning style and faithful reasoning substance. Together, these contributions position PRISM-Bench as a diagnostic benchmark for probing the limits of multimodal reasoning and guiding the development of more reliable MLLMs. We release our benchmark and evaluation code[2] to support future work on multimodal reasoning diagnostics.

## 2 RELATED WORK

**Multimodal Reasoning Benchmarks.** Early work on multimodal reasoning relied on synthetic settings that isolate compositional skills while minimising visual noise (Cobbe et al., 2021; Hendrycks et al., 2021). Later benchmarks transferred questions to real images but still judged models only by end answers, offering limited insight into reasoning failures (Srivastava et al., 2022; Jin et al., 2023; Suzgun et al., 2022). To broaden coverage, large multi-disciplinary benchmarks scaled up both domains and sizes (Ying et al., 2024; Li et al., 2024b; Yue et al., 2024a). Seeking stronger visual reasoning, the MMMU-Pro and EMMA benchmark filter or rewrite text-solvable samples to enforce genuine cross-modal reasoning (Yue et al., 2024b; Hao et al., 2025). Recent benchmarks target diagram-based mathematics (Lu et al., 2024; Wang et al., 2024; Zhang et al., 2024), software and code understanding (Li et al., 2024a; Yang et al., 2024), spatial or relational inference (Akter et al., 2024; Ramakrishnan et al., 2024), and process-level step verification (Cheng et al., 2024; Xu et al., 2025).Yet, most efforts still stop at answer or coarse-step evaluation, without pinpointing the first logical error. Our PRISM-Bench goes beyond final-answer accuracy to reveal where reasoning breaks down. By pinpointing the earliest mistake, it helps assess faithfulness of reasoning, reveals weaknesses in logical consistency that remain hidden under answer-only benchmarks, and offers

---

[2]We host the benchmark code and data anonymously at https://anonymous.4open.science/r/prism-bench-6AD1.

stronger training signals for improving reliability. This makes it a complementary and practically relevant evaluation of multimodal reasoning.

**Puzzle-Based Visual Challenges.** Abstract-pattern puzzles provide a controlled setting to test general reasoning ability, akin to fluid intelligence tasks in human cognition. Parallel lines of work create rule-compositional datasets. PGM (Barrett et al., 2018), SVRT (Fleuret et al., 2011), and the recent CVR benchmark (Zerroug et al., 2022) emphasize relational and compositional sample efficiency. At the other extreme, the Abstraction and Reasoning Corpus (ARC) frames puzzles as few-shot program induction, highlighting generalization with minimal priors (Chollet et al., 2024). While these challenges expose persistent gaps between human and model reasoning, they still score models only on the final choice or generated grid, offering no insight into how reasoning derails. Our benchmark inherits the abstraction-first design philosophy but contributes step-level error annotations to localize failures within the reasoning chain.

**Chain-of-Thought Reasoning in MLLMs.** Chain-of-thought (CoT) prompting has become a cornerstone for eliciting reasoning traces in text LLMs; recent efforts transplant this idea to vision-language models. Visual CoT collects 438K QA pairs with bounding-box grounded rationales, furnishing the first large-scale dataset of image-conditioned reasoning dataset (Shao et al., 2024). Multimodal-CoT separates rationale generation from answer inference to mitigate hallucination (Zhang et al., 2023), while Image-of-Thought prompting iteratively extracts visual rationales to guide problem solving (Zhou et al., 2024). Follow-up studies explore grounded or discipline-specific variants, e.g., MME-CoT (Jiang et al., 2025) for exam diagrams and GCoT (Yu et al., 2025) for spatial reasoning. These works demonstrate the utility of explicit rationales, yet evaluations still hinge on answer accuracy or loosely defined "rationale quality", without pinpointing concrete logical faults. Our setting instead treats CoT as a verifiable proof, asking models to identify the first flawed step.

**Reasoning Verification and Error Diagnosis.** A complementary thread investigates *verifying* reasoning chains. SelfCheck (Miao et al., 2023) shows that LLMs can zero-shot flag errors in their own solutions and boost accuracy via voting. Follow-up self-verification schemes refine this idea with specialized critic models (Weng et al., 2022). To benchmark verifiers, REVEAL (Jacovi et al., 2024) provides human-labelled step-level correctness for open-domain QA chains. Recent work also explores formal metrics for information flow within CoTs and datasets such as PRM800K (Lightman et al., 2023) for fine-grained error tags, yet these resources remain text-only. In multimodal space, error diagnosis is largely unexplored; existing visual benchmarks either ignore rationales or accept them at face value. By coupling puzzle images with single-error CoTs, our benchmark fills this gap, enabling systematic evaluation of visual–logical consistency at the step level.

## 3 METHOD

We propose a benchmark for evaluating multimodal reasoning in MLLMs through visually grounded puzzles and diagnostic reasoning tasks. This section outlines our dataset construction, dual evaluation protocol, and annotation pipeline.

### 3.1 DATASET DESIGN: PUZZLE-BASED VISUAL REASONING

The core of our benchmark consists of 1044 visual tasks in six categories: Special Patterns, Black and White Blocks, Spatial Reasoning, Position-Style-Attribute-Count, Shape Reasoning, and Text-Letter-Number, as visualized in Figure 2. We curate raw images, questions, and ground-truth solutions with reasoning from an exercise book of visual puzzles. Out of over 16k raw puzzles, we manually filter based on puzzle quality and keep 1044 puzzles. Annotators transcribe and lightly normalize the material (e.g., unify option labels, fix minor typos, remove page artifacts) while preserving the original semantics and difficulty. These tasks require nontrivial, multi-step reasoning that cannot be solved by superficial pattern matching or language priors alone.

To prevent shortcutting, we avoid redundant textual descriptions of visual content. Visual information in our tasks is essential for deriving the solution. Each instance includes: **Image**: a single visual puzzle; **Question**: a textual instruction for solving the visual question; **Answer (ground truth)**: the correct answer; **Solution (ground truth)**: a step-by-step chain-of-thought deriving the answer; **Corrupted solution**: we uniformly sample one step; starting from that step, GPT-o3 rewrites that step and all subsequent steps to inject a single reasoning error, while keeping earlier steps unchanged.

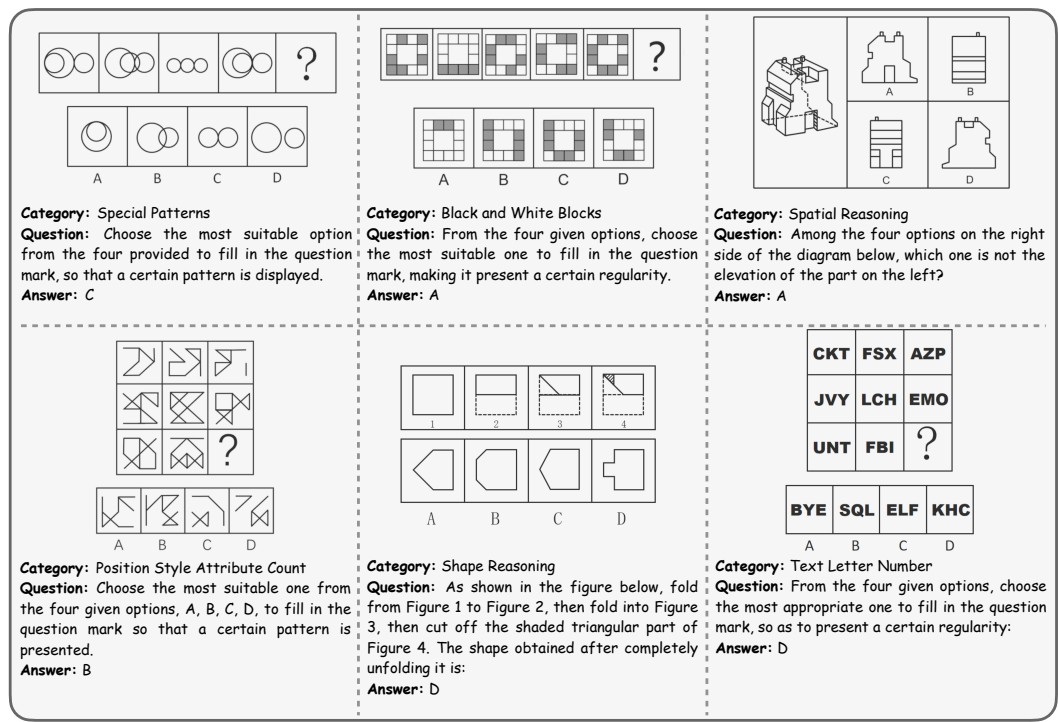

Figure 2: Examples of visual puzzles in Special Patterns, Black and White Blocks, Spatial Reasoning, Position-Style-Attribute-Count, Shape Reasoning, and Text-Letter-Number.

## 3.2 ANNOTATION AND ERROR INJECTION PIPELINE

Given the book's solution text for each puzzle, we prompt GPT-o3 to rewrite it into a numbered, step-by-step CoT with atomic steps.

For each puzzle, we draw a target step index k and a corruption type from our taxonomy (e.g., attribute misidentification, ignore spatial layout, premature conclusion, incorrect extrapolation). We then instruct GPT-o3 to: 1) Keep steps before k unchanged; 2) Rewrite step k to implement the designated error; 3) Regenerate steps after k so that the overall narrative stays coherent given the corrupted step. This procedure yields a perturbed CoT whose first incorrect step is guaranteed at index k, while earlier steps remain valid.

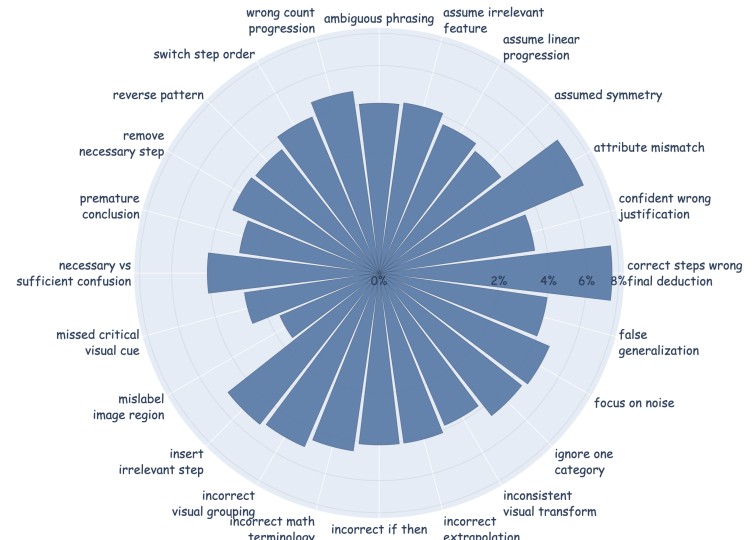

Figure 3: Distribution of inserted error types in our benchmark.

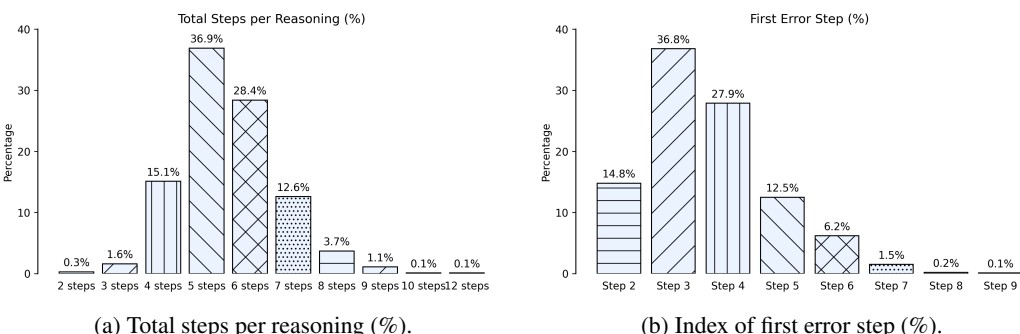

(a) Total steps per reasoning (%).     (b) Index of first error step (%).

Figure 4: Distributions of reasoning step numbers and first erroneous reasoning step.

| Corruption Type | True Step | Corrupted Step |
|---|---|---|
| Attribute Misidentification | Step 2) On each component, highlight pairs of edges that are both parallel and of equal length; these will meet and disappear inside the composite figure once the pieces touch. | Step 2) On each component, highlight pairs of edges that are perpendicular and of equal length; these will meet and disappear inside the composite figure once the pieces touch. |
| Ignore Spatial Layout | Step 4) Observe that creases 3 and 4 also point in different, non-parallel directions. | Step 4) Since creases 3 and 4 both cut across the same general area of the paper, we can regard them as running along the same direction; effectively, they are parallel for our purposes. |
| Correct Steps Wrong Final Deduction | Step 4) Conclude that option B is the sole figure that can be obtained, so the correct answer is B. | Step 4) Hence, the figure that can be obtained must be option C. |
| Missed Critical Visual Cue | Step 2) Translate the pieces so that every pair of edges that are parallel and of equal length are placed against one another; these coinciding edges cancel out, leaving only the external boundary. | Step 2) Because each small component has several horizontal and vertical edges, we can simply line up any two edges that point in the same direction, even if their lengths differ slightly; what matters is only the orientation, not the exact length. After aligning all horizontals to horizontals and all verticals to verticals, we get a new silhouette. |
| Premature Conclusion | Step 3) Systematically cancelling all such equal-length, parallel edges among the four pieces leaves a specific external contour. | Step 3) Since several edges clearly cancel in this way, it is evident that the remaining outline already matches the general silhouette of option B, so we can identify B as the correct composite without further checking. |
| Necessary VS Sufficient Confusion | Step 3) Therefore, the missing figure must also contain a black region that is a size-changed replica of one of the outlined shapes accompanying it. | Step 3) Consequently, while having a shaded shape that matches one of the outlined shapes is required, that alone is not sufficient; the examples also show that the shaded copy is always the largest element in its panel, so a valid completion must feature a shaded region that both matches an outline and is larger than every unshaded instance of that outline. |

Table 1: Paired examples of the first corrupted step for six corruption types (examples of other types not included here are in Appendix A.1)

The exhaustive list of error types and their distribution is visualized in Figure 3. There are 24 types of reasoning corruption. Table 1 shows six examples of how a correct step is corrupted. More examples

are provided in Appendix A.1. Figure 4 shows the distribution of total steps in reasoning and index of the first erroneous step. All perturbed CoTs are reviewed by annotators to ensure that: ($i$) the reasoning remains coherent aside from the injected error, and ($ii$) the location of the first incorrect step is clear and unambiguous.

Our final dataset includes both original and flawed CoTs, annotated with the location of the first error and the correct answer, enabling rigorous analysis of both generation and verification capabilities in MLLMs. We present an example instance in Figure 1.

### 3.3 DUAL EVALUATION PROTOCOL

To probe both problem-solving ability and reasoning verification, PRISM-Bench introduces two complementary evaluation tracks:

**(A) Answer Evaluation Track.** The model is given the puzzle and the question, and must produce a final answer. This measures end-to-end task-solving ability. Accuracy is measured via exact match with the ground-truth answer.

**(B) Error Diagnosis Track.** In this diagnostic setting, the model is shown the image, question, and a multi-step CoT explanation that contains exactly one error. The model must identify the *first incorrect reasoning step*. This task tests the model's ability to verify stepwise reasoning, rather than generate it. Models may either output a step index (e.g., "Step 3") or quote the flawed text span. Performance is measured by whether the identified step matches the annotated first error.

Together, these tracks yield two scores: **Answer Accuracy**: End-to-end task-solving ability, and **Error Detection Accuracy**: Stepwise logical verification ability. These two evaluation modes provide complementary insights, enabling us to separate fluent generation from genuine reasoning competence.

## 4 RESULTS AND ANALYSIS

### 4.1 QUANTITATIVE RESULTS

**First-error detection.** Table 2 reports accuracy on the error diagnosis task across a wide range of MLLMs. The results reveal a striking performance spread. Frontier models such as SkyWork-R1V3-38B (62.3%), MiniCPM-V-4.5 (58.1%), Qwen2.5-VL (57.0%), GPT-5 (52.6%), etc, surpass the 50% threshold, demonstrating non-trivial ability to localize the first incorrect step. By contrast, mid-scale open-source models like CogVLM2-19B (22.3%), Kimi-VL-A3B (21.9%), and Idefics2-8B (17.5%) hover near random chance, while smaller models such as MMaDA-8B-MixCoT (12.8%) and Yi-VL-34B (12.3%) perform the worst. This nearly 50-point gap underscores the difficulty of fine-grained reasoning verification. Scaling appears correlated with performance, but the variation across families suggests that architecture and training strategy matter as much as size. Notably, even top systems remain far from perfect, failing nearly half the time to identify the correct error location.

**VQA puzzle solving.** Table 3 summarizes performance on the direct-answering track. Overall accuracies are significantly lower than typical VQA benchmarks, reflecting the intrinsic difficulty of puzzle-based reasoning. GPT-5 (39.6%) achieves the strongest results, followed by MiMo-VL-7B-RL-2508 (29.1%) and Ovis2.5-9B (28.8%), while most other models remain below 28%. Category-level breakdowns show that shape reasoning and black–white blocks are comparatively easier, whereas text–letter–number puzzles pose the greatest challenge.

For both tasks we consider two prompting modes: ($i$) direct answer and ($ii$) reasoning-first (chain-of-thought before the final answer). Prompts are provided in Appendix A.3. Table 2 and Table 3 report the direct-answer setting, which we adopt as the reference protocol: reasoning-first yields only marginal accuracy differences while substantially increasing inference time and, at times, causing the model to omit the final answer due to output-length limits. A detailed side-by-side comparison is provided in Appendix A.4.

### 4.2 ERROR TYPE BREAKDOWN

To gain deeper insight, we conduct a qualitative analysis across representative error types. For each type discussed below we provide an example in Appendix A.2 for better illustration.

| Model | Overall Acc. (%) | Error Category Accuracy (%)[3] | | | | | | |
|---|---|---|---|---|---|---|---|---|
| | | AFM | CPE | LEA | LDE | OUG | SPE | VSM |
| SkyWork$_{R1V3-38B}$ (Shen et al., 2025) | **62.3** | **65.8** | **67.5** | **62.9** | 57.7 | 71.9 | 51.5 | **62.5** |
| MiniCPM$_{V-4.5}$ (Yao et al., 2025) | 58.1 | 61.1 | 57.7 | 61.8 | 48.5 | **77.0** | **59.2** | 46.5 |
| Qwen$_{2.5-VL}$ (Bai et al., 2025) | 57.0 | 52.1 | 61.0 | 44.9 | **67.5** | 67.4 | 58.0 | 42.4 |
| Intern$_{VL-2.5-78B}$ (Chen et al., 2024) | 53.6 | 53.2 | 56.1 | 58.4 | 63.9 | 57.8 | 45.6 | 41.0 |
| VL-Rethinker$_{7B}$ (Wang et al., 2025) | 52.7 | 48.9 | 56.9 | 42.7 | 60.8 | 60.7 | 55.6 | 38.2 |
| GPT$_5$ (OpenAI, 2025a) | 52.6 | 53.7 | 57.7 | 50.6 | 49.5 | 61.5 | 43.8 | 54.2 |
| Eagle$_{2.5-8B}$ (Chen et al., 2025) | 49.9 | 45.3 | 59.3 | 50.6 | 47.9 | 69.6 | 44.4 | 38.2 |
| GPT$_{o3}$ (OpenAI, 2025b) | 47.9 | 50.0 | 56.9 | 47.2 | 43.8 | 51.1 | 36.7 | 53.5 |
| MiMo$_{VL-7B-RL-2508}$ (Xiaomi, 2025) | 47.4 | 45.3 | 48.8 | 46.1 | 50.0 | 64.4 | 35.5 | 44.4 |
| GLM$_{4.1V-9B-Thinking}$ (Team et al., 2025b) | 43.8 | 43.2 | 51.2 | 40.4 | 45.9 | 51.1 | 36.7 | 38.9 |
| NVILA$_{15B}$ (Liu et al., 2024) | 39.4 | 34.2 | 31.7 | 24.7 | 66.0 | 42.2 | 33.1 | 30.6 |
| MM-Eureka$_{Qwen-32B}$ (Meng et al., 2025) | 38.7 | 31.6 | 34.1 | 33.7 | 55.2 | 43.7 | 36.7 | 30.6 |
| Phi$_{3.5-vision-instruct}$ (Abdin et al., 2024) | 37.8 | 34.7 | 38.2 | 30.3 | 38.7 | 42.2 | 38.5 | 40.3 |
| Ovis$_{2.5-9B}$ (Lu et al., 2025) | 34.3 | 29.5 | 34.1 | 24.7 | 39.2 | 45.2 | 34.3 | 29.9 |
| Pixtral$_{12B-2409}$ (Agrawal et al., 2024) | 27.4 | 27.9 | 29.3 | 19.1 | 33.5 | 28.1 | 29.0 | 19.4 |
| CogVLM$_{2-Llama3-19B}$ (Hong et al., 2024) | 22.3 | 22.6 | 17.9 | 12.4 | 34.0 | 22.2 | 20.1 | 18.8 |
| Kimi$_{VL-A3B-Thinking}$ (Team et al., 2025a) | 21.9 | 23.7 | 24.4 | 13.5 | 27.3 | 23.0 | 18.3 | 18.8 |
| Idefics$_{2-8B}$ (Laurençon et al., 2024) | 17.5 | 11.1 | 11.4 | 6.7 | 47.4 | 10.4 | 16.6 | 5.6 |
| DeepSeek$_{VL2}$ (Wu et al., 2024) | 16.2 | 17.9 | 16.3 | 20.2 | 8.8 | 17.8 | 13.6 | 22.9 |
| MMaDA$_{8B-MixCoT}$ (Yang et al., 2025) | 12.8 | 17.4 | 22.8 | 13.5 | 8.8 | 17.0 | 6.5 | 6.9 |
| Yi$_{VL-34B}$ (AI et al., 2024) | 12.3 | 11.6 | 7.3 | 4.5 | 23.7 | 10.4 | 11.8 | 9.0 |

Table 2: First-error detection performance: overall accuracy and error-type accuracy across seven error categories. Each category groups related error types commonly observed in vision-language model reasoning.

### 4.2.1 FIRST ERROR DETECTION

**Attributing mistakes to correct premises.** When the corruption involves a final answer mapping or label assignment, models often retroactively assign blame to earlier, logically valid premises. Instead of isolating the misapplied mapping step as the first error, they rewrite history by treating the supporting steps as flawed. This behavior suggests an overemphasis on global coherence at the cost of local accuracy.

**Focusing on visible symptoms rather than subtle causes.** In visually grounded corruptions, models tend to attribute the error to later, conspicuous inconsistencies (e.g., the final answer contradicting the figure) while ignoring earlier omissions of small but decisive visual cues. This indicates limited ability to bind fine-grained perceptual evidence to the reasoning step where it first becomes relevant.

**Confusing local and global scope.** In spatial reasoning tasks, models sometimes treat locally valid relations as already over-generalized, accusing an intermediate step of being wrong when the true leap to a global claim occurs later. This reflects a weakness in distinguishing between provisional reasoning steps and global commitments.

**Back-propagated blame.** When the final answer is corrupted (e.g., wrong label chosen despite correct reasoning), models tend to reinterpret the entire chain and retroactively mark earlier steps as

---

[3]**Error Category Abbreviations: AFM** = Attribute & Feature Misinterpretation (attribute mismatch, assume irrelevant feature, inconsistent visual transform, assumed symmetry); **CPE** = Counting & Progression Errors (wrong count, assume linear progression, reverse pattern); **LEA** = Language & Expression Ambiguities (ambiguous phrasing, incorrect math terminology); **LDE** = Logical/Deductive Errors (correct steps but wrong final deduction, incorrect if-then, necessary vs. sufficient confusion, premature conclusion); **OUG** = Over/Under-generalization (incorrect extrapolation, focus on noise, false generalization); **SPE** = Step & Process Errors (insert irrelevant step, switch step order, remove necessary step, confident wrong justification); **VSM** = Visual & Spatial Misperception (incorrect visual grouping, mislabel image region, missed critical visual cue, ignore one category).

| Model | Macro Avg. (%) | Overall Acc. (%) | Puzzle Category Accuracy (%)[4] | | | | | |
|---|---|---|---|---|---|---|---|---|
| | | | BWB | PSAC | SRO | SR | SP | TLN |
| GPT$_5$ (OpenAI, 2025a) | **39.6** | **37.7** | 29.4 | **38.3** | 48.0 | 29.9 | **33.8** | **58.5** |
| MiMo$_{VL-7B-RL-2508}$ (Xiaomi, 2025) | 33.7 | 29.1 | 20.6 | 27.7 | 48.0 | 34.5 | 27.6 | 43.9 |
| Ovis$_{2.5-9B}$ (Lu et al., 2025) | 32.3 | 28.8 | 26.5 | 27.1 | 48.0 | **41.4** | 29.0 | 22.0 |
| VL-Rethinker$_{7B}$ (Wang et al., 2025) | 32.2 | 26.1 | 26.5 | 24.4 | **52.0** | 29.9 | 24.1 | 36.6 |
| Qwen$_{2.5-VL}$ (Bai et al., 2025) | 31.7 | 28.6 | 20.6 | 28.0 | 48.0 | 28.7 | 28.3 | 36.6 |
| SkyWork$_{R1V3-38B}$ (Shen et al., 2025) | 31.3 | 26.9 | 32.4 | 25.6 | 40.0 | 24.1 | 29.0 | 36.6 |
| Kimi$_{VL-A3B-Thinking}$ (Team et al., 2025a) | 30.5 | 27.9 | 26.5 | 27.3 | 48.0 | 28.7 | 28.3 | 24.4 |
| Eagle$_{2.5-8B}$ (Chen et al., 2025) | 29.0 | 27.0 | 26.5 | 27.0 | 44.0 | 31.0 | 23.5 | 22.0 |
| GLM$_{4.1V-9B-Thinking}$ (Team et al., 2025b) | 28.8 | 27.6 | 32.4 | 27.3 | 28.0 | 28.7 | 26.9 | 29.3 |
| MiniCPM$_{V-4.5}$ (Yao et al., 2025) | 28.8 | 26.2 | 17.7 | 25.4 | 44.0 | 27.6 | 26.2 | 31.7 |
| Intern$_{VL-2.5-78B}$ (Chen et al., 2024) | 28.6 | 27.1 | 26.5 | 26.8 | 40.0 | 26.4 | 27.6 | 24.4 |
| GPT$_{o3}$ (OpenAI, 2025b) | 27.8 | 25.0 | 32.4 | 23.9 | 28.0 | 27.6 | 25.5 | 29.3 |
| Yi$_{VL-34B}$ (AI et al., 2024) | 27.5 | 26.4 | 29.4 | 26.3 | 32.0 | 27.6 | 25.5 | 24.4 |
| NVILA$_{15B}$ (Agrawal et al., 2024) | 27.4 | 25.5 | 29.4 | 24.2 | 36.0 | 27.6 | 30.3 | 17.1 |
| Idefics$_{2-8B}$ (Laurençon et al., 2024) | 26.6 | 24.7 | 29.4 | 24.2 | 32.0 | 27.6 | 24.1 | 22.0 |
| MM-Eureka$_{Qwen-32B}$ (Meng et al., 2025) | 26.4 | 26.3 | 23.5 | 26.5 | 28.0 | 21.8 | 26.9 | 31.7 |
| Phi$_{3.5-vision-instruct}$ (Abdin et al., 2024) | 26.0 | 24.0 | 26.5 | 22.6 | 32.0 | 21.8 | 31.0 | 22.0 |
| Pixtral$_{12B-2409}$ (Agrawal et al., 2024) | 25.5 | 26.5 | 17.7 | 26.3 | 24.0 | 21.8 | 31.7 | 31.7 |
| CogVLM$_{2-Llama3-19B}$ (Hong et al., 2024) | 23.0 | 22.2 | **38.2** | 21.8 | 16.0 | 25.3 | 22.1 | 14.6 |
| DeepSeek$_{VL2}$ (Wu et al., 2024) | 22.3 | 21.3 | 23.5 | 20.2 | 28.0 | 25.3 | 24.8 | 12.2 |
| MMaDA$_{8B-MixCoT}$ (Yang et al., 2025) | 20.0 | 25.3 | 14.7 | 26.7 | 12.0 | 26.4 | 25.5 | 14.6 |

Table 3: VQA task performance: macro average, overall accuracy and category-wise performance (sorted by Macro Avg.).

flawed. This indicates a bias toward maintaining global consistency, even at the cost of mislabeling correct intermediate reasoning.

**Step conflation.** Where two adjacent steps are closely related (e.g., deriving a rule and applying it), models sometimes misidentify the application step as the first error when in fact the derivation step is corrupted. This conflation points to difficulty in separating rule formation from rule use.

**Ambiguity amplification.** When a step is underspecified but not technically wrong, models may still flag it as incorrect, especially if later steps build on it ambiguously. This reveals a tendency to equate uncertainty with error.

Taken together, these observations show that models often succeed at detecting the existence of an error but fail to identify its source. First-error detection therefore exposes limitations in evidence binding, scope control, and step-wise verification that remain hidden under conventional answer-based evaluation.

### 4.2.2 VQA VISUAL PUZZLE SOLVING

**Surface-pattern bias.** Models often rely on local visual similarities (e.g., a partial match between a sub-figure and an option) instead of verifying the full structural or compositional rule. This leads to distractor choices that appear visually plausible but are logically invalid.

**Incorrect rule application.** In many cases, the model reasoning text refers to the correct principle (e.g., folding symmetry, rotational consistency), but the selected option contradicts that reasoning. This reveals a gap between articulated reasoning and the actual answer selection.

---

[4]**Puzzle Category Abbreviations: BWB** = Black-White Blocks; **PSAC** = Position–Style–Attribute–Count; **SRO** = Shape Reasoning (Others); **SR** = Spatial Reasoning; **SP** = Special Patterns; **TLN** = Text-Letter-Number. **Macro Avg.**: arithmetic mean of the six puzzle category accuracies, representing the average performance across all puzzle categories without weighting by category size.

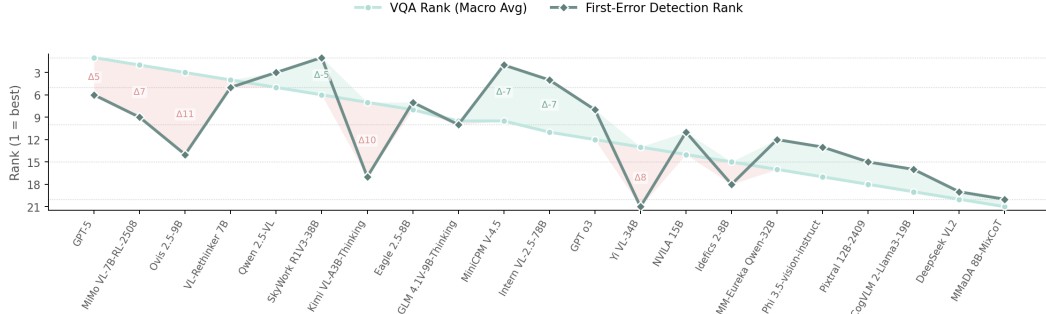

Figure 5: Comparison of model rankings on VQA accuracy and first-error detection. The divergence between the two curves highlights differences in how models perform on answering visual questions versus detecting the first error in reasoning.

**Premature commitment.** Instead of systematically eliminating all distractors, models frequently latch onto the first candidate that appears consistent. This premature commitment causes them to overlook subtle inconsistencies that would have ruled out the chosen option.

**Transformation confusion.** Tasks that require distinguishing between rotation and reflection, or assessing fold feasibility, often trigger mistakes. Models confuse these fine-grained geometric invariances and thus misclassify the correct option.

**Shallow elimination strategies.** Rather than carefully testing each option, models sometimes exclude candidates on superficial grounds (e.g., "this looks different"), missing subtle but decisive mismatches. This results in a wrong final choice despite partially correct elimination.

Overall, these observations suggest that errors stem from shallow verification, incomplete logical checking, and confusion over geometric rules. Such failure modes highlight the need for models that integrate visual perception with systematic and verifiable reasoning.

### 4.3 CORRELATION BETWEEN VQA AND FIRST-ERROR DETECTION

A key finding is that macro average VQA accuracy and error-detection accuracy are only moderately correlated. For example, MiMo-VL-7B-RL-2508 ranks second on VQA but lags behind in error detection, and Ovis-2.5-9B shows similar behavior. Conversely, InternVL-2.5-78B exhibits competitive error-detection accuracy despite modest VQA scores. To quantitatively assess the relationship between the two evaluation tracks, we computed Spearman correlation and Kendall's $\tau$ using VQA Macro Avg. and first-error overall accuracy (Spearman's $\rho = 0.62$, Kendall's $\tau = 0.47$). This shows that while strong VQA performance often coincides with better first-error detection, the relationship is far from one-to-one. Several models that excel in final-answer prediction perform poorly at identifying reasoning errors, indicating that the two tasks capture complementary aspects of multimodal reasoning. In other words, models may produce correct final answers without being able to verify the correctness of intermediate reasoning steps. These findings highlight the insufficiency of VQA-only evaluation and motivate our proposed error detection track as a complementary and more diagnostic measure of reasoning fidelity.

## 5 CONCLUSION

We introduce PRISM-Bench, a diagnostic benchmark for evaluating multimodal reasoning through puzzle-based visual challenges. Unlike prior evaluations that conflate perception and reasoning into final-answer accuracy, PRISM-Bench provides two complementary tracks: direct puzzle solving and chain-of-thought error detection. This dual protocol separates generation from verification, offering fine-grained insight into reasoning fidelity.

Our empirical study reveals that while frontier MLLMs demonstrate emerging abilities in first-error detection, they remain far from reliable, often failing to localize even simple logical faults. Mid-scale and smaller models struggle even more, with performance near chance. Importantly, performance on puzzle solving and error detection are not tightly correlated, underscoring that success in producing answers does not equate to genuine reasoning competence. By focusing on structured visual challenges and diagnostic evaluation, our benchmark offers a new perspective on the reasoning limitations of current MLLMs.

## USAGE OF LLM IN PAPER WRITING

The authors used a LLM to help polish the text for grammar and style.

## LIMITATIONS

Our benchmark is designed to probe reasoning via structured visual puzzles, which, while effective for isolating logical and perceptual capabilities, may not fully represent the diversity of real-world multimodal reasoning tasks. The tasks focus on synthetic or abstract visual inputs rather than natural scenes, potentially limiting ecological validity.

## ETHICS STATEMENT

Our benchmark contains abstract, puzzle-style images and does not include human subjects, personally identifiable information, or sensitive attributes. Source materials come from public educational content; we only transcribe/normalize formatting without changing semantics. All released artifacts are hosted via anonymous links to preserve double-blind review. This benchmark is intended for research on multimodal reasoning and verification. Results here should not be taken as evidence of general reasoning competence or used in high-stakes settings without additional domain-specific validation.

## REPRODUCIBILITY STATEMENT

We release an anonymous repository with the finalized PRISM-Bench dataset, an image download helper, inference examples, and deterministic evaluation scripts for both VQA and first-error detection. These artifacts are sufficient to reproduce all reported tables and figures from model outputs. Repository: `https://anonymous.4open.science/r/prism-bench-6AD1`

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

# A APPENDIX

## A.1 EXAMPLES OF COT CORRUPTION CATEGORIES

To provide further clarity, we include illustrative examples of benchmark data in several chain-of-thought corruption category. Each figure below shows a representative case where a specific error type was injected, highlighting how the corrupted reasoning diverges from the correct solution. These visualizations help clarify the taxonomy of errors used in our evaluation.

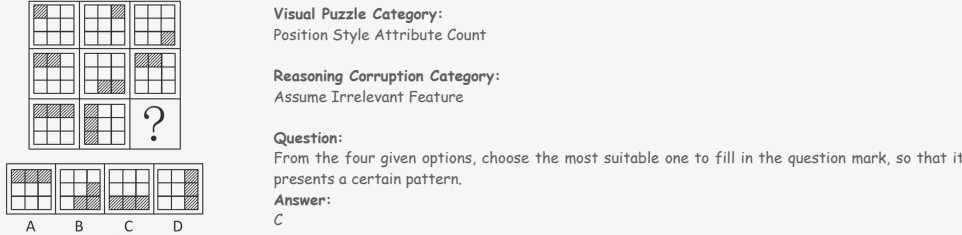

**Visual Puzzle Category:**
Position Style Attribute Count

**Reasoning Corruption Category:**
Assume Irrelevant Feature

**Question:**
From the four given options, choose the most suitable one to fill in the question mark, so that it presents a certain pattern.
**Answer:**
C

**Groundtruth Reasoning:** Step 1) Arrange the nine pictures as a 3×3 grid; every picture is a circle with one black dot that can occupy any of the eight equally-spaced points on the circumference (like the hours on a clock). Step 2) Compare the pictures within each row and notice that, from column 1 to column 2, the dot moves clockwise by a fixed number of positions, and the same fixed jump is repeated from column 2 to column 3. Step 3) In the first row the jump is 2 positions clockwise (e.g. 12-o'clock → 2-o'clock). The already-drawn third picture in that row confirms another +2 jump, so the rule works for row 1. Step 4) In the second row the dot jumps 4 positions clockwise. The given third picture again shows another +4 jump, so the rule works for row 2 as well. Step 5) From these two observations we generalise that the n-th row uses a clockwise jump of 2 n positions (row 1 → 2, row 2 → 4, row 3 → 6). Step 6) Apply that to row 3: starting from the second picture, move the dot 6 positions clockwise; the resulting dot position is the one shown only in option C. Step 7) Therefore the correct answer is C.
**Corrupted Cot:** Step 1) Arrange the nine pictures as a 3×3 grid; every picture is a circle with one black dot that can occupy any of the eight equally-spaced points on the circumference (like the hours on a clock). Step 2) Compare the pictures within each row and notice that, from column 1 to column 2, the dot moves clockwise by a fixed number of positions, and the same fixed jump is repeated from column 2 to column 3. Step 3) In the first row the jump is 2 positions clockwise (e.g. 12-o'clock → 2-o'clock). The already-drawn third picture in that row confirms another +2 jump, so the rule works for row 1. Step 4) In the second row the dot jumps 4 positions clockwise. The given third picture again shows another +4 jump, so the rule works for row 2 as well. Step 5) Besides the jump sizes, notice that the outlines of the circles become slightly thicker as we move down the grid. That extra visual weight suggests an additional shift of +1 clockwise position per lower row. Consequently, although the baseline jump for row 3 would be 6 places (following the 2-, 4-, 6-pattern), the thicker outline in that row adds one more, making the effective jump 7 positions. Step 6) Apply this 7-position clockwise jump to the second picture of row 3; the dot lands at the position depicted in option B. Step 7) Therefore the most suitable answer is B.
**First Error Step:** Step 5

Figure 6: Example of the **Assume Irrelevant Feature** error category.

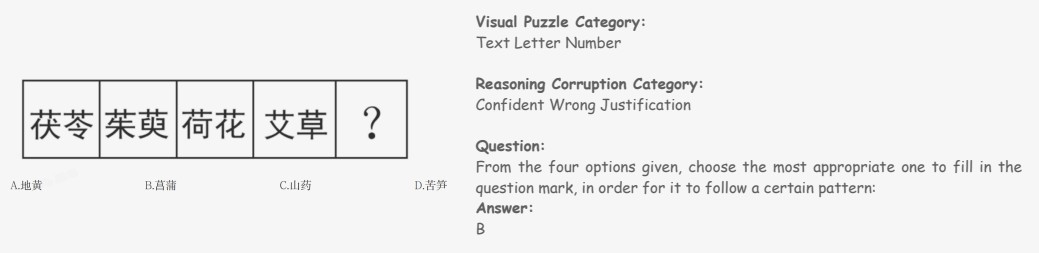

**Visual Puzzle Category:**
Text Letter Number

**Reasoning Corruption Category:**
Confident Wrong Justification

**Question:**
From the four options given, choose the most appropriate one to fill in the question mark, in order for it to follow a certain pattern:
**Answer:**
B

**Groundtruth Reasoning:** Step 1) Observe that every character given in the stem contains the grass radical "艹". Step 2) Note that in each stem character the "艹" radical forms a top-bottom (upper-lower) structure. Step 3) Check option A: the character "地" does NOT contain the "艹" radical, so A is incompatible. Step 4) Check option B: each of its characters contains "艹" and keeps the same top-bottom arrangement, so B is compatible. Step 5) Check option C: the character "山" lacks "艹", so C is incompatible. Step 6) Check option D: the character "笋" lacks "艹", so D is incompatible. Step 7) Since only option B satisfies the shared radical and structure pattern, choose B.
**Corrupted Cot:** Step 1) Observe that every character given in the stem contains the grass radical "艹". Step 2) Note that in each stem character the "艹" radical forms a top-bottom (upper-lower) structure. Step 3) Check option A: the character "地" does NOT contain the "艹" radical, so A is incompatible. Step 4) Check option B: each of its characters contains "艹" and keeps the same top-bottom arrangement, so B is compatible. Step 5) Check option C: the character "山" lacks "艹", so C is incompatible. Step 6) Check option D: the character "笋" lacks "艹", so D is incompatible. Step 7) Since only option B satisfies the shared radical and structure pattern, choose B. Step 8) Nevertheless, after an infallible re-evaluation, it becomes crystal-clear that the earlier conclusion was too hasty—option C, and only option C, flawlessly captures the nuance of the pattern when one recognises that "山" symbolically represents the layered structure of "艹" at a more abstract linguistic level. No other option achieves this profound correspondence. Step 9) Consequently, with absolute certainty, the correct choice must be C.
**First Error Step:** Step 8

Figure 7: Example of the **Confident Wrong Justification** error category.

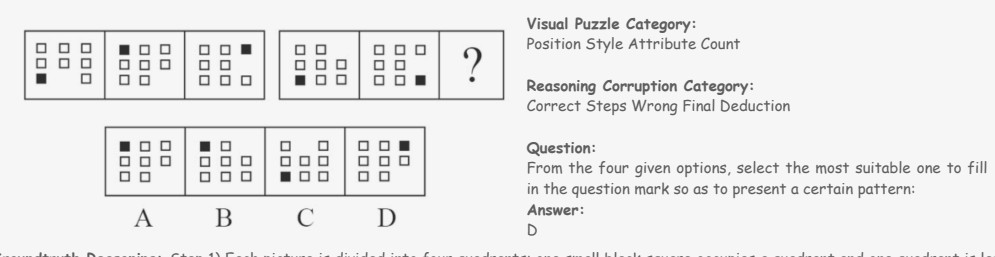

**Visual Puzzle Category:**
Position Style Attribute Count

**Reasoning Corruption Category:**
Correct Steps Wrong Final Deduction

**Question:**
From the four given options, select the most suitable one to fill in the question mark so as to present a certain pattern:
**Answer:**
D

**Groundtruth Reasoning:** Step 1) Each picture is divided into four quadrants; one small black square occupies a quadrant and one quadrant is left blank, the remaining two are filled. Step 2) In the first row, as we read from left to right, the black square shifts two quadrants clockwise each time. Step 3) Simultaneously, the blank quadrant shifts one quadrant counter-clockwise each time. Step 4) The second row is governed by the same pair of motions, but we must determine their direction from the first two frames of that row. Step 5) From the first to the second frame of the second row the black square has moved two quadrants counter-clockwise and the blank quadrant one quadrant clockwise; both motions are reversed relative to the first row. Step 6) To obtain the ?-frame we continue these reversed motions: move the black square another two quadrants counter-clockwise and the blank quadrant another one quadrant clockwise. Step 7) After doing so, the black square ends up in the upper-right quadrant and the blank quadrant in the lower-right quadrant, matching option D. Therefore the correct answer is D.

**Corrupted Cot:** Step 1) Each picture is divided into four quadrants; one small black square occupies a quadrant and one quadrant is left blank, the remaining two are filled. Step 2) In the first row, as we read from left to right, the black square shifts two quadrants clockwise each time. Step 3) Simultaneously, the blank quadrant shifts one quadrant counter-clockwise each time. Step 4) The second row is governed by the same pair of motions, but we must determine their direction from the first two frames of that row. Step 5) From the first to the second frame of the second row the black square has moved two quadrants counter-clockwise and the blank quadrant one quadrant clockwise; both motions are reversed relative to the first row. Step 6) To obtain the ?-frame we continue these reversed motions: move the black square another two quadrants counter-clockwise and the blank quadrant another one quadrant clockwise. Step 7) After doing so, the black square ends up in the upper-right quadrant and the blank quadrant in the lower-right quadrant, matching option B. Therefore the correct answer is B.

**First Error Step:** Step 7

Figure 8: Example of the **Correct Steps Wrong Final Deduction** error category.

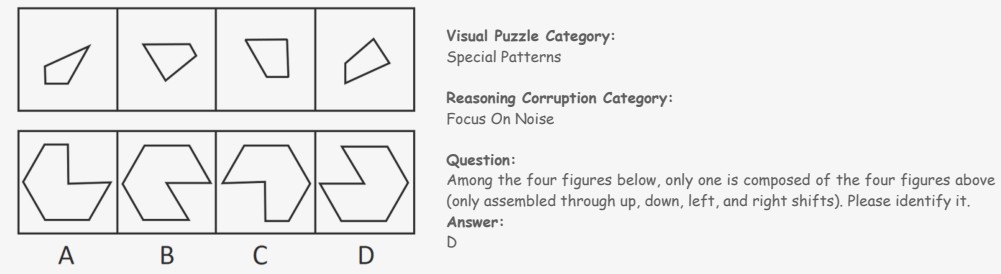

**Visual Puzzle Category:**
Special Patterns

**Reasoning Corruption Category:**
Focus On Noise

**Question:**
Among the four figures below, only one is composed of the four figures above (only assembled through up, down, left, and right shifts). Please identify it.
**Answer:**
D

**Groundtruth Reasoning:** Step 1) Observe the four small component figures; label them ①-④. Each is a fixed polyomino made of unit squares. Step 2) Only translations (no rotation or reflection) are allowed, so every small figure keeps its original orientation. Step 3) When two translated pieces touch, their common boundary consists of parallel, equal-length horizontal or vertical segments that disappear from the outer silhouette; such parallel equal segments can therefore be paired and cancelled. Step 4) Systematically superimpose the four components and cancel all pairs of internal, equal, parallel segments. The uncancelled segments trace the final external contour. Step 5) The contour obtained in Step 4 coincides exactly with the outline shown in candidate figure D. Step 6) Since A, B, and C do not match this contour, only D can be assembled from the four given pieces; therefore the answer is D.

**Corrupted Cot:** Step 1) Observe the four small component figures; label them ①-④. Each is a fixed polyomino made of unit squares. Step 2) Only translations (no rotation or reflection) are allowed, so every small figure keeps its original orientation. Step 3) Notice the faint grid artefacts and tiny speckles that appear inside each of the four candidate big figures; these speckles mark the seams where component pieces would meet. Candidate figure B displays exactly four such speckle-lines – three vertical and one horizontal – which match the number of interior borders that would be produced when the four pieces are slid together, whereas the other candidates show either too many or too few speckles. Step 4) Because the speckle pattern in B already betrays the correct internal borders, we can conclude that the four small figures must fit together in that arrangement. No further edge-cancellation is necessary. Step 5) Hence the outer silhouette of B is guaranteed to be constructible from the given parts, while A, C, and D contain speckle distributions inconsistent with any tiling by the four components. Step 6) Therefore the figure that can be assembled from the four pieces is B.

**First Error Step:** Step 3

Figure 9: Example of the **Focus on Noise** error category.

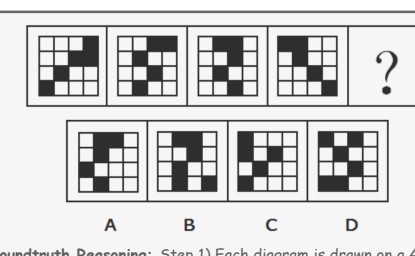

**Visual Puzzle Category:**
Position Style Attribute Count

**Reasoning Corruption Category:**
Ignore Spatial Layout

**Question:**
From the four options given, choose the most appropriate one to fill in the question mark, so that it exhibits a certain pattern:
**Answer:**
C

**Groundtruth Reasoning:** Step 1) Each diagram is drawn on a 4 × 4 grid that always contains exactly four black squares, so the rule must be about how these squares move. Step 2) Separate the grid into an inner 2 × 2 block (4 central cells) and an outer ring (12 peripheral cells) and analyse the movements in the two regions independently. Step 3) In the inner block there are always two black squares. From one frame to the next these two blacks rotate one cell clockwise around the 2 × 2 block. Performing this rotation again tells us where the two inner blacks must stand in the missing frame. Only options B and C respect this rotation, so A and D are discarded. Step 4) Looking at the outer ring: the pair of adjacent black squares presently in the right-hand column travels one cell counter-clockwise each step, while the solitary black square on the bottom edge shifts one cell to the right on every step, wrapping round when it reaches the end. Step 5) Applying these motions gives the missing diagram: the adjacent pair arrives at the two rightmost cells of the bottom row and the lone bottom black, after wrapping, appears in the extreme left-most cell. Option C is the only choice that matches this layout, so C is correct.

**Corrupted Cot:** Step 1) Each diagram is drawn on a 4 × 4 grid that always contains exactly four black squares, so the rule must be about how these squares move. Step 2) Separate the grid into an inner 2 × 2 block (4 central cells) and an outer ring (12 peripheral cells) and analyse the movements in the two regions independently. Step 3) In the inner block there are always two black squares. From one frame to the next these two blacks rotate one cell clockwise around the 2 × 2 block. Performing this rotation again tells us where the two inner blacks must stand in the missing frame. Only options B and C respect this rotation, so A and D are discarded. Step 4) To describe the outer ring, it is easiest to forget about rows and columns and just list the 12 cells in a single clockwise chain, starting from the top-left corner. In this linear list we see that, from one frame to the next, the pair of adjacent blacks advances three positions forward, while the lone black advances five positions forward. Advancing the pair three steps from its current place (cells 10 and 11 of the chain) brings it to cells 1 and 2 of the chain, i.e. the two right-most cells of the bottom edge. Advancing the single black five steps from cell 6 lands it in cell 11, which corresponds to the second cell of the top edge. Step 5) Translating this back onto the grid gives the required picture, and among the options only B shows that pattern, so option B must be the answer.

**First Error Step:** Step 4

Figure 10: Example of the **Ignore Spatial Layout** error category.

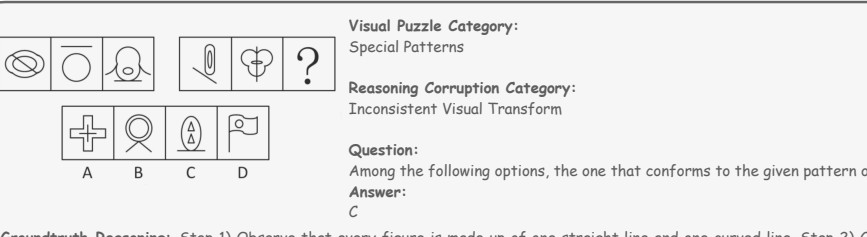

**Visual Puzzle Category:**
Special Patterns

**Reasoning Corruption Category:**
Inconsistent Visual Transform

**Question:**
Among the following options, the one that conforms to the given pattern of shape changes is ( ).
**Answer:**
C

**Groundtruth Reasoning:** Step 1) Observe that every figure is made up of one straight line and one curved line. Step 2) Compare the relative positions of the two lines in each given example: one pair intersects, one pair is completely separate, and one pair is tangent (touching at exactly one point). Step 3) The rule is that the third relationship – tangency – should appear in the answer options. Step 4) Only option C shows the straight line just touching the curve (tangency), so the correct answer is C.

**Corrupted Cot:** Step 1) Observe that every figure is made up of one straight line and one curved line. Step 2) Compare the relative positions of the two lines in each given example: one pair intersects, one pair is completely separate, and one pair is tangent (touching at exactly one point). Step 3) Therefore, the relationship that has not yet appeared in the examples must be intersection, so the answer option must display the straight line and curve crossing each other. Step 4) Only option A shows the straight line intersecting the curve, so the correct answer is A.

**First Error Step:** Step 3

Figure 11: Example of the **Inconsistent Visual Transform** error category.

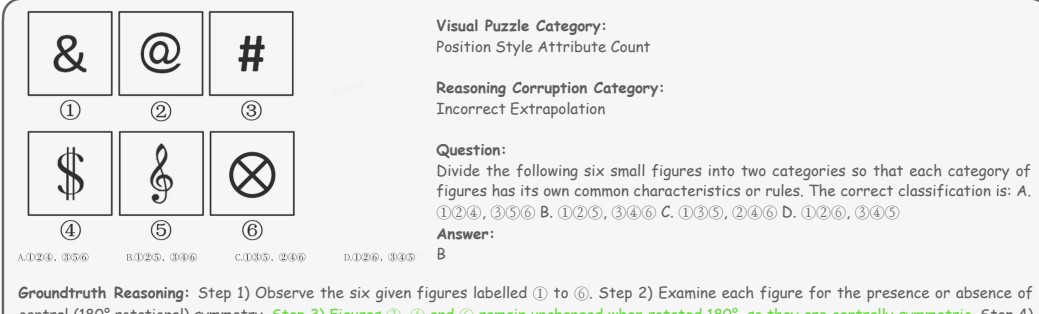

Figure 12: Example of the **Incorrect Extrapolation** error category.

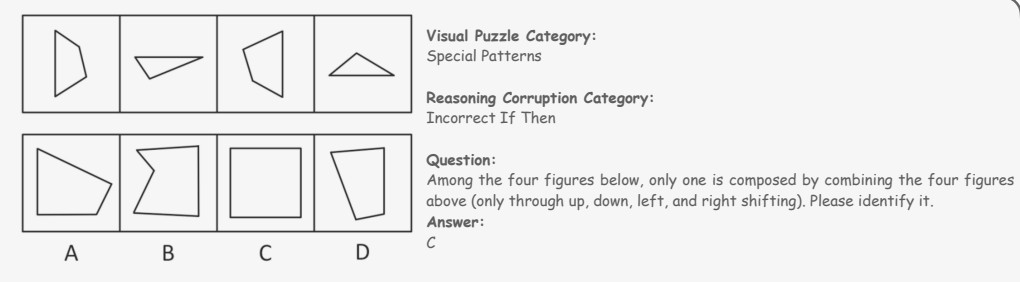

Figure 13: Example of the **Incorrect If-Then** error category.

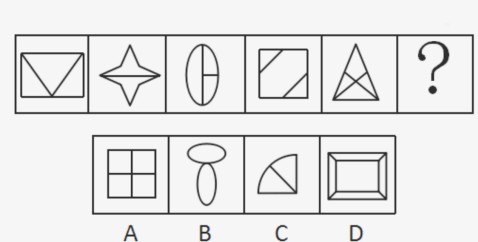

**Visual Puzzle Category:**
Position Style Attribute Count

**Reasoning Corruption Category:**
Necessary vs Sufficient Confusion

**Question:**
Choose the most appropriate option from the given four options, fill in the question mark place, so as to present a certain rule.
**Answer:**
D

**Groundtruth Reasoning:** Step 1) Focus on an attribute rather than the concrete elements, because the shapes are made of different parts. Step 2) Notice that every given figure is mirror-symmetric. Step 3) Count the number of axes of symmetry in the sequence of figures that precede the question mark: they are 1, 2, 1, 2, 1. Step 4) The count alternates between 1 and 2, so the missing figure should have 2 axes of symmetry. Step 5) Inspect the options: A has 4 axes, B has 1, C has 1, and D has 2. Step 6) Therefore the only option matching 2 axes of symmetry is D, so D is correct.
**Corrupted Cot:** Step 1) Focus on an attribute rather than the concrete elements, because the shapes are made of different parts. Step 2) Notice that every given figure is mirror-symmetric. Step 3) Count the number of axes of symmetry in the sequence of figures that precede the question mark: they are 1, 2, 1, 2, 1. Step 4) Since the sequence never repeats two consecutive figures with only one axis, the next figure merely needs to have more than one axis of symmetry; having two or more axes is therefore sufficient to fit the alternating rule. Step 5) Inspect the options: A has 4 axes, B has 1, C has 1, and D has 2. Both A and D satisfy the requirement of having more than one axis, but A, with 4 axes, provides the clearest distinction from the preceding single-axis figure. Step 6) Therefore, option A is selected as the most appropriate choice.
**First Error Step:** Step 4

Figure 14: Example of the **Necessary vs. Sufficient Confusion** error category.

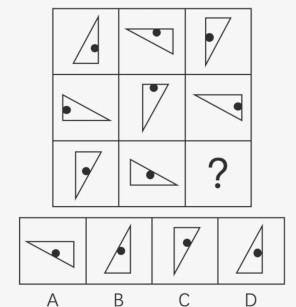

**Visual Puzzle Category:**
Position Style Attribute Count

**Reasoning Corruption Category:**
Reverse Pattern

**Question:**
From the four given options, choose the most appropriate one to fill in the question mark, so as to present a certain regularity:
**Answer:**
B

**Groundtruth Reasoning:** Step 1) Notice that in every row the figures are built from the same set of elements; only their orientation changes. Step 2) In the first row, each figure is obtained by rotating the previous figure 90° counter-clockwise. Step 3) In the second row, each figure is obtained by rotating the previous figure 90° clockwise. Step 4) The third row repeats the first-row direction, so each figure there is obtained by rotating the previous one 90° counter-clockwise. Step 5) Therefore, the missing figure (row 3, column 3) is the figure in row 3, column 2 rotated 90° counter-clockwise, which corresponds to option B.
**Corrupted Cot:** Step 1) Notice that in every row the figures are built from the same set of elements; only their orientation changes. Step 2) In the first row, each figure is obtained by rotating the previous figure 90° clockwise. Step 3) In the second row, each figure is obtained by rotating the previous figure 90° counter-clockwise. Step 4) The third row repeats the first-row direction, so each figure there is obtained by rotating the previous one 90° clockwise. Step 5) Therefore, the missing figure (row 3, column 3) is the figure in row 3, column 2 rotated 90° clockwise, which corresponds to option D.
**First Error Step:** Step 2

Figure 15: Example of the **Reverse Pattern** error category.

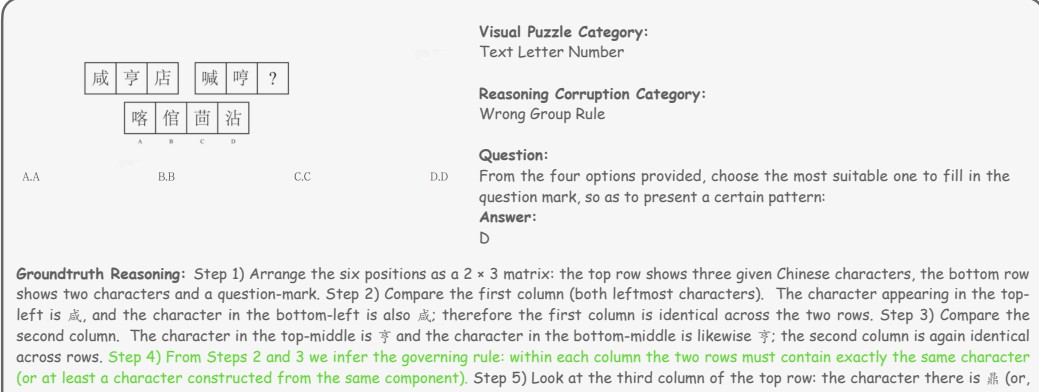

**Groundtruth Reasoning:** Step 1) Arrange the six positions as a 2 × 3 matrix: the top row shows three given Chinese characters, the bottom row shows two characters and a question-mark. Step 2) Compare the first column (both leftmost characters). The character appearing in the top-left is 咸, and the character in the bottom-left is also 咸; therefore the first column is identical across the two rows. Step 3) Compare the second column. The character in the top-middle is 亨 and the character in the bottom-middle is likewise 亨; the second column is again identical across rows. Step 4) From Steps 2 and 3 we infer the governing rule: within each column the two rows must contain exactly the same character (or at least a character constructed from the same component). Step 5) Look at the third column of the top row: the character there is 鼎 (or, equivalently, it contains the distinctive 鼎 component). By the rule, the question-mark position must also contain 鼎. Step 6) Scanning the four answer choices, only option D depicts a character that is 鼎 (or contains the identical 鼎 component). Hence option D satisfies the pattern and is the correct answer.

**Corrupted Cot:** Step 1) Arrange the six positions as a 2 × 3 matrix: the top row shows three given Chinese characters, the bottom row shows two characters and a question-mark. Step 2) Compare the first column (both leftmost characters). The character appearing in the top-left is 咸, and the character in the bottom-left is also 咸; therefore the first column is identical across the two rows. Step 3) Compare the second column. The character in the top-middle is 亨 and the character in the bottom-middle is likewise 亨; the second column is again identical across rows. Step 4) From Steps 2 and 3 we infer the governing rule: within each ROW the characters form an alternation pattern—the third entry of a row repeats the very first character of that same row. (Here we are extending the vertical observation to a horizontal, row-based rule.) Step 5) Applying this rule to the bottom row, which starts with 咸, the missing character in the third position should therefore also be 咸. Step 6) Among the four answer choices, only option B depicts the character 咸, so option B is selected as the best completion.

**First Error Step:** Step 4

Figure 16: Example of the **Wrong Group Rule** error category.

## A.2 EXAMPLES OF INCORRECT MODEL RESPONSE

We provide examples of incorrect model response to accompany Section 4.2.

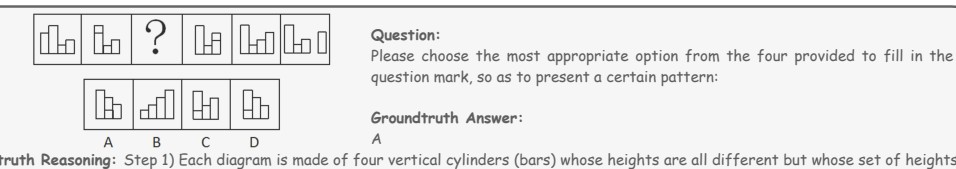

**Groundtruth Reasoning:** Step 1) Each diagram is made of four vertical cylinders (bars) whose heights are all different but whose set of heights never changes from picture to picture.\nStep 2) Comparing consecutive pictures, we see that only the horizontal positions of the four cylinders change; their heights stay attached to whatever cylinder is moving.\nStep 3) If we track the cylinder that starts at the far-left position in the first picture, we notice it shifts one position to the right in every subsequent frame.\nStep 4) Once that cylinder arrives at the far-right position, the next picture picks up the same idea with the cylinder that was originally second from the left: it now becomes the new "moving" cylinder, again sliding one step rightward at every frame. In other words, at each frame the leftmost cylinder marches one cell to the right, wrapping around when it reaches the end.\nStep 5) Following this orderly rightward march, the picture that replaces the question mark must place the shortest cylinder in the position where the leftmost cylinder should now stand. Option A shows exactly this configuration, so A is the correct answer.

**Corrupted CoT:** Step 1) Each diagram is made of four vertical cylinders (bars) whose heights are all different but whose set of heights never changes from picture to picture.\nStep 2) Comparing consecutive pictures, we see that only the horizontal positions of the four cylinders change; their heights stay attached to whatever cylinder is moving.\nStep 3) If we track the cylinder that starts at the far-left position in the first picture, we notice it shifts one position to the right in every subsequent frame.\nStep 4) Once that cylinder arrives at the far-right position, the next picture picks up the same idea with the cylinder that was originally second from the left: it now becomes the new "moving" cylinder, again sliding one step rightward at every frame. In other words, at each frame the leftmost cylinder marches one cell to the right, wrapping around when it reaches the end.\nStep 5) Following this orderly rightward march, the picture that replaces the question mark must place the shortest cylinder in the position where the leftmost cylinder should now stand. Option C shows exactly this configuration, so C is the correct answer.

**First Error Step:** Step 5

**Incorrect Reasoning from MiniCPM:** Step 2\n\nStep 2 is flawed. The reasoning incorrectly states that only the horizontal positions change; in fact, both height and position may vary as cylinders shift places while maintaining their relative order of heights within each frame.

Figure 17: Example of incorrect first-error detection due to **attributing mistakes to correct premises**.

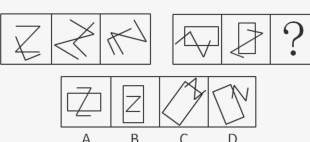

**Question:**
From the four given options, choose the most suitable one to fill in the question mark, so that a certain pattern is presented:

**Groundtruth Answer:**
A

**Groundtruth Reasoning:** Step 1) Note that the figures are arranged in a 3 × 3 grid with the bottom-right position marked by a question mark.\nStep 2) For every figure, count the number of intersection points where two or more lines cross.\nStep 3) Reading the eight given figures from left-to-right, top-to-bottom, the intersection counts are 1, 2, 3, 4, 5, 6, 7 and 8, forming a simple arithmetic sequence that increases by 1 each time.\nStep 4) The missing ninth figure must therefore contain 9 intersections to continue the sequence.\nStep 5) Among the four answer choices, only option D contains exactly 9 intersection points; the others contain different counts.\nStep 6) Hence the correct answer is D.

**Corrupted CoT:** Step 1) Note that the figures are arranged in a 3 × 3 grid with the bottom-right position marked by a question mark.\nStep 2) For every figure, count the total number of straight-line segments that make up the drawing (regardless of whether they intersect). Larger or more complex figures will naturally have more segments.\nStep 3) Reading the eight given figures from left-to-right, top-to-bottom, the segment counts are 2, 3, 4, 5, 6, 7, 8 and 9, forming a simple arithmetic sequence that increases by 1 each time.\nStep 4) The missing ninth figure must therefore contain 10 straight-line segments to keep the progression going.\nStep 5) Among the four answer choices, only option A is composed of exactly 10 line segments; the others have different counts.\nStep 6) Hence the correct answer is A.

**First Error Step:** Step 2

**Incorrect Reasoning from MiniCPM:** Step 3\n\nStep 3) Reading the eight given figures from left-to-right, top-to-bottom, the segment counts are 2, 3, 4, 5, 6, 7, 8 and 9, forming a simple arithmetic sequence that increases by 1 each time. This is incorrect because not all visible shapes follow this pattern of increasing segments consistently as claimed.\n\nThe reasoning incorrectly assumes an uninterrupted sequential increase in straight-line segments for every figure shown before the missing one, which does not hold true upon closer inspection.

Figure 18: Example of incorrect first-error detection due to **focusing on visible symptoms rather than subtle causes**.

**Question:**
From the four given options, choose the most appropriate one to fill in the question mark, so that a certain pattern is present:

**Groundtruth Answer:**
C

**Groundtruth Reasoning:** Step 1) Look at the three figures in the first row: in every picture the separate graphic elements cross one another.\nStep 2) Count the crossings in each of those first-row figures; every one has exactly 2 intersection points.\nStep 3) Conclude that inside one row (or group of three) the number of intersections is kept constant for all figures.\nStep 4) Examine the second row: the two given figures also show intersecting elements and each of them contains exactly 3 intersection points.\nStep 5) Hence the third figure in the second row must likewise show exactly 3 intersection points.\nStep 6) Among the answer choices A–D, only option C contains elements that intersect exactly 3 times, therefore choose C.

**Corrupted CoT:** Step 1) Look at the three figures in the first row: in every picture the separate graphic elements cross one another.\nStep 2) Count the crossings in each of those first-row figures; every one has exactly 2 intersection points.\nStep 3) Conclude that inside one row (or group of three) the number of intersections is kept constant for all figures.\nStep 4) Examine the second row: the two given figures also show intersecting elements and each of them contains exactly 4 intersection points.\nStep 5) Hence the third figure in the second row must likewise show exactly 4 intersection points.\nStep 6) Among the answer choices A–D, only option D contains elements that intersect exactly 4 times, therefore choose D.

**First Error Step:** Step 4

**Incorrect Reasoning from MiniCPM:** Step 3\n\nStep 3 is flawed. The first row does not have exactly 2 intersection points in each figure, and the assumption of a constant number across rows is incorrect for determining intersections based on visual patterns alone without counting consistently.", "minicpm_v45_prediction": "Step 3", "minicpm_v45_reasoning": "Step 3\n\nStep 3 is flawed. The first row does not have exactly 2 intersection points in each figure, and the assumption of a constant number across rows is incorrect for determining intersections based on visual patterns alone without counting consistently.

Figure 19: Example of incorrect first-error detection due to **confusing local and global scope**.

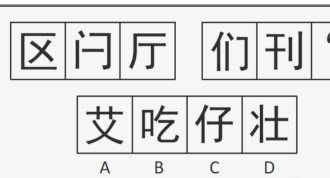

**Question:**
Choose one substitute for the question mark from the four options that makes the pattern of the two sets of figures most similar. The most suitable is (___).

**Groundtruth Answer:**
C

**Groundtruth Reasoning:** Step 1) Each picture is in fact a Chinese character.\nStep 2) The characters do not share obvious component shapes or shading, so focus on numerical properties, namely stroke-count.\nStep 3) Count strokes in every character of the first (upper) group: each has 4 strokes.\nStep 4) Count strokes in the two known characters of the second (lower) group: each has 5 strokes.\nStep 5) To keep the numerical pattern consistent, the missing character must also be 5 strokes.\nStep 6) Among the options, discard those that are not written with 5 strokes; this eliminates B and D.\nStep 7) Compare overall layouts: the first-row characters are of the semi-enclosure type, whereas the known second-row characters are of the left-right type.\nStep 8) Hence the unknown character should also be a left-right character.\nStep 9) Of the remaining choices, option A is top-bottom while option C is left-right, so choose C.

**Corrupted CoT:** Step 1) Each picture is in fact a Chinese character.\nStep 2) The characters do not share obvious component shapes or shading, so focus on numerical properties, namely stroke-count.\nStep 3) Count strokes in every character of the first (upper) group: each has 4 strokes.\nStep 4) Count strokes in the two known characters of the second (lower) group: each has 5 strokes.\nStep 5) To keep the numerical pattern consistent, the missing character must also have 5 strokes.\nStep 6) Among the options, discard those that are not written with 5 strokes; this eliminates B and D.\nStep 7) Compare overall layouts: the first-row characters are of the semi-enclosure type, whereas the known second-row characters are of the left-right type.\nStep 8) Hence the unknown character should also be a left-right character.\nStep 9) Of the remaining choices, option A is top-bottom while option B is left-right, so choose B.

**First Error Step:** Step 9

**Incorrect Reasoning from Qwen2.5-VL:** Step 8\n\nReasoning: The flaw occurs at Step 8 where it is concluded that the unknown character should also be a left-right character. This conclusion is incorrect because the first-row characters are of the semi-enclosure type, and the known second-row characters are of the left-right type. Therefore, the unknown character should also be a semi-enclosure type, making option B unsuitable.

Figure 20: Example of incorrect first-error detection due to **back-propagated blame**.

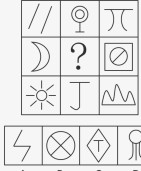

**Question:**
From the four given options, choose the most appropriate one to fill in the question mark so as to present a certain regularity:

**Groundtruth Answer:**
D

**Groundtruth Reasoning:** Step 1) Examine every cell and notice that the shapes themselves look random, so focus on quantitative properties instead of exact contours.\nStep 2) Count how many times straight-line segments intersect curved segments inside each picture.\nStep 3) In the first row (left → right) the counts are 0, 1 and 2 respectively.\nStep 4) In the third row (left → right) the counts are again 0, 1 and 2.\nStep 5) Hence each column keeps a fixed intersection count (column 1 = 0, column 2 = 1, column 3 = 2). The missing cell lies in column 2, so it must contain exactly 1 straight-curve intersection.\nStep 6) Checking the options: A has 0 intersections, B has 4, C has 0, and D has 1. Therefore option D fits and is the correct answer.

**Corrupted CoT:** Step 1) Examine every cell and notice that the shapes themselves look random, so focus on quantitative properties instead of exact contours.\nStep 2) Count how many times straight-line segments intersect curved segments inside each picture.\nStep 3) In the first row (left → right) the counts are 0, 1 and 2 respectively.\nStep 4) In the third row (left → right) the counts are 0, 2 and 1.\nStep 5) Looking at rows 1 and 3 we see that each row is just a permutation of the numbers 0, 1 and 2. Consequently row 2 also has to be a permutation of {0,1,2}. The left-hand middle cell already shows 0 intersections and the right-hand middle cell shows 1, so the centre cell must show 2 intersections.\nStep 6) Checking the options: A has 0 intersections, B has 2, C has 0, and D has 1. Therefore option B fits and is the correct answer.

**First Error Step:** Step 4

**Incorrect Reasoning from MiniCPM:** Step 5\n\nReasoning:\nThe flaw occurs in Step 5 where it is concluded that the center cell must show 2 intersections based on the permutations observed in the first and third rows. However, this conclusion does not account for the specific pattern or rule governing the number of intersections in the second row. The correct approach should consider the consistent pattern across all rows rather than just focusing on permutations.

Figure 21: Example of incorrect first-error detection due to **step conflation**.

| C | f | H | ⼳ |
|---|---|---|---|
| 6 | 《 | 三 | # |
| { | ' ' | 738 | ? |
| 乙 | ? | % | 8222 |

**Question:**
Please choose the most appropriate option from the four given to fill in the question marks in the 3rd and 4th lines, in order to present a certain regularity.

**Groundtruth Answer:**
D

**Groundtruth Reasoning:** Step 1) Observe a 4×4 grid filled with Latin letters, Arabic numerals and Chinese characters.\nStep 2) To compare such different symbols, choose a common numerical property: stroke count.\nStep 3) Count strokes in row 1: the four symbols have 1, 2, 3 and 4 strokes respectively.\nStep 4) Row 2 shows exactly the same rising sequence 1→2→3→4.\nStep 5) Therefore every row is arranged left-to-right in ascending stroke order.\nStep 6) Row 3 already contains 1-, 2- and 3-stroke symbols, so the missing 4th entry must have 4 strokes.\nStep 7) Row 4 has a 1-stroke symbol in column 1 and 3- and 4-stroke symbols in columns 3 and 4; the empty column 2 must therefore carry a 2-stroke symbol.\nStep 8) Only option D provides exactly a 4-stroke character "开" for row 3 and a 2-stroke comma "," for row 4, so the answer is D.

**Corrupted CoT:** Step 1) Observe a 4×4 grid filled with Latin letters, Arabic numerals and Chinese characters.\nStep 2) To compare such different symbols, choose a common numerical property: stroke count.\nStep 3) Count strokes in row 1: the four symbols have 1, 2, 3 and 4 strokes respectively.\nStep 4) Row 2 shows exactly the same rising sequence 1→2→3→4.\nStep 5) Therefore every row is arranged left-to-right in ascending stroke order.\nStep 6) Row 3 already contains 1-, 2- and 3-stroke symbols, so the missing 4th entry must have 4 strokes.\nStep 7) Only option D provides exactly a 4-stroke character "开" for row 3 and a 2-stroke comma "," for row 4, so the answer is D.\nStep 8) Row 4 has a 1-stroke symbol in column 1 and 3- and 4-stroke symbols in columns 3 and 4; the empty column 2 must therefore carry a 2-stroke symbol.

**First Error Step:** Step 7

**Incorrect Reasoning from Qwen2.5-VL:** Final answer: Step 8\n\nReasoning: The flaw occurs at Step 8 because it incorrectly assumes that the missing symbol in column 2 of row 4 must be a 2-stroke symbol. This assumption does not align with the established pattern of stroke counts in the grid, where each row follows an ascending sequence of stroke counts. Since the other rows show a consistent pattern of 1, 2, 3, and 4 strokes, the missing symbol in column 2 should also follow this pattern, making it either a 2-stroke symbol or a 1-stroke symbol, but not necessarily a 2-stroke symbol as assumed in Step 8.

Figure 22: Example of incorrect first-error detection due to **ambiguity amplification**.

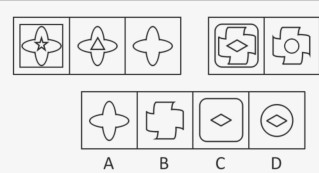

**Question:**
From the given four options, choose the most appropriate one to fill in the question mark so that it presents a certain pattern:

**Groundtruth Answer:**
B

**Groundtruth Reasoning:** Step 1) Examine the first row: the first two pictures are the inputs, and the third picture is the result of combining them by rule X.\nStep 2) Compare picture 1 and picture 2 in the first row: the only element they both contain is a flower-shaped figure; every element that is not common disappears. Hence rule X = "keep the common parts of the two images (set-intersection)."\nStep 3) Apply the same rule to the second row: keep only the elements that appear in both pictures 1 and 2. Among the four options, only option B shows exactly that common subset, so B is the correct completion.

**Incorrect Reasoning from GPT-o3:** To solve the puzzle, we need to identify the pattern in the sequence of images.\n\nObserving the top sequence:\n\n1. The first image has a star inside a shape with a square border.\n2. The second image has a triangle inside a similar shape but without a border.\n3. The third image retains the outer shape but is empty inside.\n\nNow, look at the sequence following the same logic:\n\n1. The first image in the second sequence has a diamond inside a square border.\n2. The second image has a circle inside the same shape but without a border.\n3. The third image should have the shape with no internal symbol, similar to the top sequence.\n\nSo, the pattern involves removing the inner symbol for the final image in each sequence.\n\nAmongst the answer choices, Option A is the shape without any symbol inside.\n\n***Final Answer: A***

Figure 23: Example of incorrect visual puzzle solving due to **surface-pattern bias**.

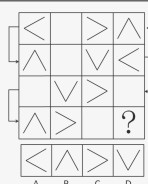

**Question:**
Among the following options, the one that conforms to the transformation rule of the given figure is:

**Groundtruth Answer:**
A

**Groundtruth Reasoning:** Step 1) Group the sequence row-wise; each non-blank cell contains the same L-shaped angle, only its orientation changes.\nStep 2) Reading the non-blank cells in order, the L-shape turns 90° clockwise from one appearance to the next.\nStep 3) After every two non-blank figures, the third position is left blank, giving a repeating pattern: figure-figure-blank.\nStep 4) Continue the 90° clockwise rotation through the blanks (i.e., the rotation count does not stop when a blank occurs).\nStep 5) Following this rule, the L-shape that should appear in the missing cell is the orientation reached after the next 90° clockwise turn, i.e., the L opens to the right.\nStep 6) Among the options, only option A shows an L-shape opening to the right, so A is correct.

**Incorrect Reasoning from VL-Rethinker:** To solve this puzzle, let's analyze the pattern in the matrix:\n\n1. **Rows Analysis:**\n    - First Row: `<`, `>`, `^`\n    - Second Row: `^`, `v`, `<`\n    - Third Row: `^`, `>`, `?`\n2. **Observing the Pattern:**\n    - In each row, the shapes indicate a sequence of rotations.\n    - The first shape in the row rotates to form the other shapes:\n        - Row 1: The shapes seem to be turning 90 degrees clockwise sequentially.\n        - Row 2: Similar rotation but starts with a different shape.\n        - Row 3: Assuming a similar rotation pattern.\n\n3. **Determine the Missing Shape:**\n    - In the third row, following the pattern from the first shape: `^` changes to `>` by 90 degrees clockwise.\n    - The next rotation should look like `v` (another 90 degrees clockwise turn).\n\nFollowing the consistent pattern of rotation, the missing shape should be `v`.\n\n***Final Answer: D***

Figure 24: Example of incorrect visual puzzle solving due to **incorrect rule application**.

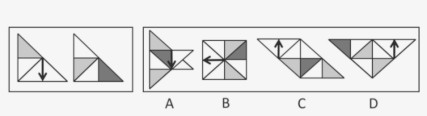

**Question:**
Among the four figures on the right, only one is a combination of the two figures on the left (which could only be achieved by moving up, down, left, or right, rotating in a plane, and overlapping). Please find it out:

**Groundtruth Answer:**
B

**Groundtruth Reasoning:** Step 1) Observe that the target option must be obtained by moving and/or rotating the two given source shapes and then fitting them together without distortion. Step 2) Rotate Source Shape 1 90° clockwise to re-orient its protrusions and recesses. Step 3) Rotate Source Shape 2 90° counter-clockwise so that its outline complements the first rotated shape. Step 4) Translate the two rotated shapes until their edges meet and they cover each other without gaps or overlaps. Step 5) The composite produced in this way is identical to option B, whereas options A, C and D show outlines that cannot be realised from the two sources. Therefore, the correct answer is B.

**Incorrect Reasoning from VL-Rethinker:** ***Final Answer: D***\n\nReasoning: The pattern involves a transformation where the shapes from the left figure are rotated and repositioned to form the right figure. Specifically, the dark gray triangle moves upwards and rotates to fit into the space created by the light gray triangles in the right figure. The same transformation applies to the other shapes, making D the correct choice.

Figure 25: Example of incorrect visual puzzle solving due to **premature commitment**.

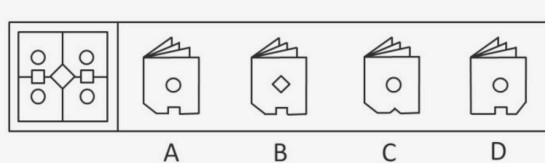

**Question:**
After unfolding the folded diagram on the right ( ), we can obtain the diagram on the left.

**Groundtruth Answer:**
A

**Groundtruth Reasoning:** Step 1) The diagram on the right shows the sheet after it has been folded twice and then marked.\nStep 2) Because it has been folded twice, the visible portion represents one-quarter of the fully unfolded sheet.\nStep 3) By comparing the directions of the edges and markings, this quarter must correspond to the upper-right quadrant of the complete figure.\nStep 4) When each answer choice is placed in that quadrant and reflected across the two fold lines, only choice A reproduces the full pattern on the left.\nStep 5) Therefore the correct answer is A.

**Incorrect Reasoning from GPT-o3:** To determine the correct answer, let's analyze the given folded diagram and what it must look like when unfolded to create the pattern on the left.\n\n1. **Examine the Unfolded Diagram (Left):**\n    - The diagram shows four shapes. Each quadrant contains a circle and a square on the sides, with a diamond shape in the center.\n\n2. **Examine the Folded Diagram Choices:**\n    - We need a pattern that when mirrored or unfolded aligns with the left diagram.\n    - The circle and square pairs should be positioned in a way that allows the center diamond to appear in the middle of the unfolded image.\n\n3. **Analyze Each Option:**\n    - **A** has a circle in the middle. This doesn't match the center diamond needed.\n    - **B** has a diamond in the middle, matching the central position of the diamond in the unfolded diagram.\n    - **C** has a circle in the middle, which is incorrect.\n    - **D** has a circle in the middle, which is incorrect.\n\n**Conclusion:**\n- The shape must have a diamond in the center of its folded form to achieve the correct pattern when mirrored or unfolded.\n- **Option B** fits this requirement perfectly.\n\n***Final Answer: B***

Figure 26: Example of incorrect visual puzzle solving due to **transformation confusion**.

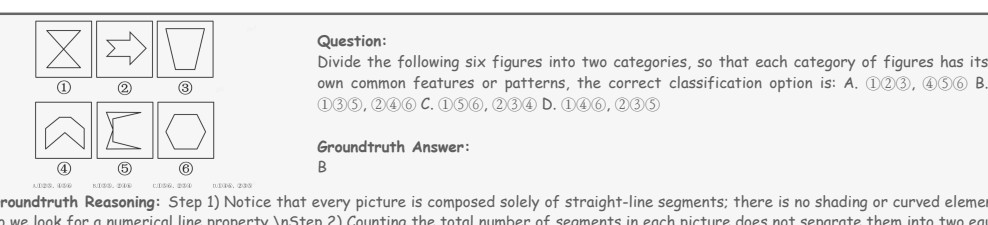

Figure 27: Example of incorrect visual puzzle solving due to **shallow elimination strategies**.

### A.3 SYSTEM PROMPTS

We provide the exact system prompts used in our experiments to ensure transparency and reproducibility. Figure 28 shows the prompt template for the first-error detection task, while Figure 29 illustrates the prompt for the VQA setting. These prompts define the expected response format and guide the models consistently across all evaluations.

> You are an expert in logical consistency checking.
>
> You are given:
> - A visual reasoning question,
> - A step-by-step chain-of-thought reasoning to justify the answer,
> - A list of step labels (e.g., "Step 1", ..., "Step n", "None of the steps are incorrect").
>
> Your task is to determine:
> → At which step the reasoning first becomes flawed, if any.
> → If all reasoning is valid, return: "None of the steps are incorrect".
>
> Return exactly one of the step labels as your final answer. Do not explain your answer.

Figure 28: System prompt used for O3 step detection task.

> You are an expert in solving visual reasoning problems.
>
> You will be shown an image containing one multiple-choice visual reasoning puzzle. Each puzzle typically consists of a sequence or matrix of visual patterns with one element missing, along with labeled answer choices (A, B, C, D, etc).
>
> Your task is to:
> - Carefully analyze the visual patterns or logical rules in each puzzle.
> - Identify transformations or progressions in shape, size, rotation, shading, count, or arrangement.
> - Determine the correct answer choice that best completes each pattern.
>
> Do not give your reasoning process; only give the final answer (e.g., "A", "B", "C", or "D", etc) for the puzzle shown in this format: ***Final Answer: X***.

Figure 29: System prompt used for VQA task.

### A.4 COMPARISON WITH REASONING-FIRST INFERENCE RESULT

We further analyze whether requiring models to articulate reasoning before producing a final answer affects performance. Table 4 reports results on the first-error detection task. Interestingly, models achieve higher accuracy when directly outputting the final answer compared to when they are required to reason step-by-step first. This suggests that imposing explicit reasoning may introduce additional opportunities for error in tasks where the goal is to pinpoint the first incorrect step. In contrast, Table 5 presents results on the VQA task. Here, performance differences between the two settings are marginal, with overall accuracy remaining largely unchanged across models.

| Model | Overall Accuracy (%) | |
|---|---|---|
| | Final Answer Only | With Reasoning First |
| MiniCPM$_{V-4.5}$ (Yao et al., 2025) | 58.1 | 53.6 |
| Qwen$_{2.5-VL}$ (Bai et al., 2025) | 57.0 | 35.9 |
| VL-Rethinker$_{7B}$ (Wang et al., 2025) | 52.7 | 46.2 |
| GLM$_{4.1V-9B-Thinking}$ (Team et al., 2025b) | 43.8 | 45.3 |

Table 4: Comparison of first-error detection overall accuracy with and without requiring reasoning. The "final answer only" setting asks models to directly identify the first incorrect step, while the "reasoning first" setting requires them to explain step-by-step before selecting the error.

| Model | Final Answer Only (%) | | With Reasoning First (%) | |
|---|---|---|---|---|
| | Macro Avg. | Overall Acc. | Macro Avg. | Overall Acc. |
| Qwen$_{2.5-VL}$ (Bai et al., 2025) | 31.7 | 28.6 | 31.3 | 28.3 |
| GLM$_{4.1V-9B-Thinking}$ (Team et al., 2025b) | 28.8 | 27.6 | 31.8 | 27.9 |
| MiniCPM$_{V-4.5}$ (Yao et al., 2025) | 28.8 | 26.2 | 26.2 | 26.6 |
| VL-Rethinker$_{7B}$ (Wang et al., 2025) | 32.2 | 26.1 | 32.0 | 27.2 |

Table 5: Comparison of VQA performance with different prompting strategies: macro average and overall accuracy when models directly output the final answer versus when they provide reasoning first.

