# OpenReview forum: "Where Did the Reasoning Go Wrong? A Benchmark of Puzzle-Based Visual Tasks with CoT Error Detection"
_ICLR.cc/2026/Conference — ICLR 2026 Conference Withdrawn Submission_

### Official Review · Reviewer_YT1f · 2025-10-16

**Soundness:** 3
**Presentation:** 3
**Contribution:** 2
**Rating:** 4
**Confidence:** 4

**Summary:**

The paper constructs a new benchmark, PRISM-Bench, for evaluation of final answer and first error detection of VQA on VLMs. The benchmark includes more than 1000 puzzles, and each puzzle has a correct CoT and a corrupted CoT with a known first error step. They evaluate a large number of off-the-shelf VLMs and demonstrate that their benchmark is still challenging for most VLMs.

**Strengths:**

- This puzzle benchmark includes 24 fine-grained error types, which covers many cases and would be useful for evaluation and future training of VLMs.
- Their evaluation on various VLMs demonstrate that the task is challenging for many frontier models, especially for the first error detection.
- They provide a lot of details for reproduction.

**Weaknesses:**

- The paper claims a dual-track protocol with both final answer accuracy and first error-detection. In fact, this is not a new thing as first error detection has been proposed in Self-Explore [1] in 2024. As the metric is not new, this dataset seems to be just another VQA puzzles.
- There is no ablation study on whether the mistakes made by VLMs are due to their recognition ability or reasoning ability. For example, the effect of recognition ability can be tested by using images of different resolution.
- The authors do not discuss how first error-detection can help general reasoning ability.


[1] SELF-EXPLORE: Enhancing Mathematical Reasoning in Language Models with Fine-grained Rewards. EMNLP 2024.

**Questions:**

What is the distribution of number of CoT steps in this benchmark?

---

### Official Review · Reviewer_FpwN · 2025-10-25

**Soundness:** 2
**Presentation:** 3
**Contribution:** 2
**Rating:** 4
**Confidence:** 5

**Summary:**

This paper introduces PRISM-Bench, a benchmark of visual puzzles aimed at evaluating not only whether multimodal large language models (MLLMs) can produce correct answers, but also how their reasoning unfolds. Unlike previous evaluations that focus solely on final-answer accuracy, PRISM-Bench introduces a diagnostic task: given a visual puzzle and a step-by-step chain of thought (CoT) containing exactly one error, a model must identify the first incorrect step. Evaluations across multiple open and proprietary MLLMs reveal that even recent models often fail to detect basic logical faults in the provided reasoning chains.

**Strengths:**

1. The paper tackles an important problem - assessing whether MLLMs can perform fact-checking and reasoning verification conditioned on visual input. This topic is of clear interest to the ICLR community.
2. PRISM-Bench offers a targeted diagnostic benchmark that shifts the focus from answer correctness to reasoning verification, an underexplored dimension of model trustworthiness.
3. The benchmark includes 1,044 puzzles with a balanced distribution across 24 corruption types, covering symbolic, geometric, and analogical reasoning.
4. The evaluation demonstrates that leading models struggle with error identification and that their capabilities in solving tasks versus detecting reasoning errors vary significantly - an interesting and underreported observation.
5. The paper is clearly written, well structured, and supported by illustrative figures and qualitative examples.
6. The literature review is comprehensive and well contextualized.

**Weaknesses:**

1. The dataset reportedly derives from “an exercise book of visual puzzles,” yet the source is not cited. The authors should explicitly reference the data source and clarify which components are derived versus newly created. In particular, how was the ground-truth step-by-step solution obtained? Was it provided in the original material or generated by the authors?
2. Section 3.2 states that the method “keeps states before k unchanged.” If the corrupted reasoning chains were generated by editing step-by-step solutions with GPT-o3 from step k onward, then the task might implicitly reduce to identifying where the text style or phrasing changes.
    1. Is GPT-o3 conditioned on earlier steps to maintain consistent style?
    2. To validate dataset integrity, the authors should add a text-only baseline (without images) that attempts to detect the corruption purely from linguistic cues.
3. Figure 4b indicates high class imbalance (e.g., 0.1% examples in “Step 9” vs. 36.8% in “Step 3”). As such, accuracy alone is not informative - a trivial classifier predicting “Step 3” achieves 36.8%, surpassing several MLLMs in Table 2. The authors should include macro F1 or balanced accuracy to account for imbalance.
4. The insights obtained from qualitative analysis in Section 4.2 are difficult to translate into practical steps. The error taxonomy is interesting but underdeveloped.
    1. What distinguishes “Attributing mistakes to correct premises” from “Back-propagated blame”?
    2. How could the insights inform future model design or prompting strategies? It would strengthen the work if the authors explored few-shot or zero-shot prompts incorporating these failure modes.
5. The study is conducted exclusively in the zero-shot setting. Exploring few-shot settings could show whether reasoning verification can be improved through in-context learning.

Minor:
1. Figure 3 would benefit from showing absolute counts in addition to percentages.
2. In Section 3.2, please report the number of annotators and their agreement (e.g., Cohen’s κ).
3. In the error diagnosis track, consider using a distance-based metric (e.g., the mean absolute error between true and predicted indices of the first incorrect step) to quantify near misses.
4. Clarify which model variants and parameter sizes were evaluated (e.g., Qwen2.5-VL has variants ranging from 3B to 72B parameters trained in different setups).
5. Including human performance would provide an important baseline.

**Questions:**

1. What is the exact source and license of the “exercise book” used to derive PRISM-Bench puzzles?
2. How are ground-truth step-by-step solutions produced? Were human annotators involved?
3. How do you ensure that GPT-o3 edits do not create detectable stylistic discontinuities?
4. Did you experiment with models receiving only text (without the visual puzzle) to test for possible linguistic shortcuts?
5. Why was accuracy chosen as the primary metric despite class imbalance?
6. Could you report human performance on the “identify the first incorrect step” task for comparison?

**Details Of Ethics Concerns:**

The dataset reportedly derives from “an exercise book of visual puzzles,” yet the source is not cited. The authors should explicitly reference the data source and clarify which components are derived versus newly created.

---

### Official Review · Reviewer_J2vh · 2025-10-26

**Soundness:** 4
**Presentation:** 3
**Contribution:** 3
**Rating:** 4
**Confidence:** 5

**Summary:**

This paper investigates the error-detection ability in logical reasoning within multimodal contexts—a capability that has been extensively explored in text-only settings but remains under-studied when visual or other modalities are involved. The authors construct a well-designed benchmark and comprehensively evaluate a broad range of state-of-the-art models, revealing a persistent gap between ideal reasoning and current performance. Their analysis further shows that error-detection ability is not always correlated with reasoning capability itself. While the overall novelty of the challenge may be somewhat limited, the problem is meaningful, and the authors’ efforts toward establishing a rigorous benchmark and systematic evaluation are valuable and timely

**Strengths:**

Great efforts must be paid the manually curating the 1044 high-quality puzzles out of the original 16 raw challenges, although the number is relatively small if the 1044 are divided into different categories of corruption types. If the structure of the benchmark should be more detailed clarified, I can more tell the reliability of the conclusions.

A wide range of state-of-the-art models are evaluated on the benchmark, which provides a strong basis for conclusions about MLLMs’ error-detection ability on logical puzzles. Nevertheless, if a more detailed explanation of the model selection criteria (e.g., open-source vs. closed-source models, and different model sizes) is provided, it will be easier to tell how representative these conclusions truly are.

**Weaknesses:**

The task itself lacks sufficient novelty. There already exist several benchmarks featuring puzzle-based visual reasoning challenges—such as VisualSphinx (https://arxiv.org/abs/2505.23977), which provides 660K puzzles—and multimodal benchmarks focused on error detection in mathematical reasoning, such as ErrorRadar (https://arxiv.org/abs/2410.04509), which is not cited in the related work. Although this paper aims to combine error-detection and logical puzzle reasoning, it is not entirely clear that building a new benchmark is necessary. If the primary goal is to study error-detection ability, the authors could have leveraged existing datasets instead of constructing a new one from textbook materials.

Although substantial effort was devoted to manually curating the dataset, the resulting number of examples (1,044) is relatively small and may be insufficient for comprehensively evaluating models’ ability to detect diverse categories of reasoning errors.

**Questions:**

•	It would be helpful to clarify which version of Qwen2.5-VL was used in the experiments—3B, 7B, or 32B. When reporting results across multiple models, specifying the model size (when available) would make the comparison clearer and more informative.
•	Why not leverage existing puzzle-based benchmarks instead of manually curating new puzzle-style VQAs? Utilizing established datasets could reduce unnecessary effort and allow the authors to focus on generating more examples of erroneous reasoning trajectories and conducting post-training explorations to improve error-detection capability.
•	For each of the 1,022 puzzle examples, how many corrupted chains of thought (CoTs) are included? Do these contain multiple categories of errors or multiple locations of the “first error”? A more detailed explanation of the benchmark data construction would greatly improve clarity and reproducibility.

---

### Official Review · Reviewer_L6GT · 2025-10-30

**Soundness:** 2
**Presentation:** 3
**Contribution:** 2
**Rating:** 4
**Confidence:** 3

**Summary:**

The paper introduces PRISM-Bench for evaluating MLLMs. Each data point contains an image, question, answer, solution, and a corrupted solution, with GPT-o3 prompted to rewrite the step-by-step CoT solution and the corrupted solution, enabling evaluation of both problem-solving performance and the ability to identify the first incorrect step in a step-by-step CoT reasoning chain.
Experiments across a wide range of MLLMs, yet primarily open-source models, show consistently low performance on both tasks and only a moderate correlation between VQA accuracy and first-error detection, highlighting a gap between fluent generations and faithful reasoning.

**Strengths:**

- The proposed benchmark supports two evaluation types: VQA and first-error detection. Performance results show that the benchmark is challenging yet differentiable across models, a good sign of its diagnostic value. The first-error detection task also fills the gap by introducing process-level reasoning evaluation into the multimodal domain.
- The paper also provides sufficient qualitative analysis, including breakdowns across error types and detailed case studies.

**Weaknesses:**

- The idea of first-error detection has been explored in text-only domains. In addition to PRM800K, works such as ProcessBench [1] are closely related and should be discussed.
- The overall construction involves using GPT-o3 to rewrite step-by-step CoT solutions and to create corrupted versions, yet the specific prompts used for these operations are not included.
- Human annotators are also involved to ensure the coherence of the reasoning and the correctness of the first incorrect step, but limited information is provided. What is the background of the annotators? How was the annotation conducted, and how was final quality ensured (through consensus or inter-annotator agreement)? Since this dataset is intended to be open-sourced, these details should be made more transparent.
- The raw data appears to be sourced from an exercise book, but this is described only vaguely. The data sources should be made more explicit, as there may be potential copyright issues.
- The evaluation could include some more frontier proprietary models, such as Claude and Gemini.

Overall, I think that as a benchmark paper, the current version lacks some necessary transparency and detail.

[1] ProcessBench: Identifying Process Errors in Mathematical Reasoning, ACL 2025

**Questions:**

- The first-error detection task seems suitable for Process Reward Models, yet the evaluation mainly involves prompting general-purpose MLLMs. Have you tried PRMs such as VisualPRM [1] on this task?
- Based on the examples, many questions involve pattern-finding, which could benefit from in-context examples. However, according to the given prompts, the main evaluation appears to be zero-shot. Have you conducted any few-shot experiments using QA pairs from the same category as demo samples for the VQA task? Given that each data point already includes both the problem and groundtruth solution (clean one), this should be feasible.

[1] VisualPRM: An Effective Process Reward Model for Multimodal Reasoning, arxiv 2025

---

### Official Review · Reviewer_uVPo · 2025-11-02

**Soundness:** 2
**Presentation:** 2
**Contribution:** 2
**Rating:** 4
**Confidence:** 2

**Summary:**

PRISM-Bench is a benchmark designed to evaluate the stepwise reasoning accuracy of multimodal large language models (MLLMs). Unlike traditional visual question answering (VQA) tasks that focus solely on final answer accuracy, PRISM-Bench employs a dual evaluation mechanism: (i) assessing final task accuracy through puzzle-solving tasks; (ii) requiring models to identify the first erroneous step in a corrupted chain of thought (CoT) through error localization tasks. The benchmark comprises 1,044 curated visual puzzles spanning six reasoning scenarios and introduces 24 distinct error types.

**Strengths:**

- Introduces first-error detection as a new evaluation axis for CoT reasoning in vision-language tasks — a significant departure from answer-only benchmarks.
- Carefully curated visual puzzles, error-injected CoTs, and 24 error types provide fine-grained diagnostic power.
- The benchmark can inform the development of more interpretable and trustworthy multimodal systems.

**Weaknesses:**

- The curation process (from 16k to 1,044 tasks) is under-detailed; selection criteria and annotator agreement metrics are missing.
- Deeper model-wise or architecture-wise breakdown could clarify which design choices aid error detection.

**Questions:**

How well do humans perform on first-error detection? Including human-level accuracy would help calibrate the task difficulty.

---

### Public Comment · ~Lei_Yang8 · 2025-11-13
**The Ground-Truth of First-Error Detection track is terrible.**

I only checked the questions our model got wrong among the first 20 prompts, and 6 of them were actually due to incorrect ground-truth labels; see PRISM-Bench-incorrect-GTs.pdf in https://drive.google.com/drive/folders/10IVvbmZNZ0Owey4cA5n9Vaz0gnaZAqpR

Moreover, the `Groundtruth Reasoning` of the examples presented in the current paper contains many hallucinations and errors, for examples:

- Figure 8: Step 1) Example of the Correct Steps Wrong Final Deduction error category
    - Line#818: Each picture is divided into **four quadrants**;
- Figure 11: Example of the Inconsistent Visual Transform error category
    - Line#910: Step 1) Observe that every figure is made up of one straight line and **one curved line**.
- Figure 13: Example of the Incorrect If-Then error category.
    - Line#960: Step 1) Examine the four given fragments: they are polyomino-like shapes made of **equal squares**; each has a distinctive profile of notches and protrusions.

Based on my analysis above, the error rate of GT could be as high as 30% (6/20), which does not align with what the paper claims in lines #271–#273:
> All perturbed CoTs are reviewed by annotators to ensure that: (i) the reasoning remains coherent aside from the injected error, and (ii) the location of the first incorrect step is clear and unambiguous

This suggests a critical bug in the first-error-detection track. Conclusions drawn from this track in its current form risk substantially misleading the community.

I comment here to help prevent interested researchers from repeating the same cycle I experienced—excitement upon seeing the first-error-detection task, shocked and disappointed after running it, and frustration after tracing the underlying GT problems—and will save everyone’s time and energy.

---

### Note · Authors · 2025-11-14

I have read and agree with the venue's withdrawal policy on behalf of myself and my co-authors.